

# Dataset variability and carbonate concentration influence the performance of local visible-near infrared spectral models

Simon Oberholzer[1], Laura Summerauer[2], Markus Steffens[1,3], Chinwe Ifejika Speranza[1]

[1]Institute of Geography, University Bern, 3012 Bern, Switzerland
[2]Department of Environmental System Science, ETH Zürich, 8092 Zürich, Switzerland
[3]Department of Soil Science, Research Institute of Organic Agriculture, 5070 Frick, Switzerland

*Correspondence to*: Laura Summerauer (laura.summerauer@usys.ethz.ch)

**Abstract.** The application of visual and near infrared soil spectroscopy (vis–NIR) is an easy and cost-efficient way to gain a wide variety of soil information to cover high spatial and temporal resolution in large-scale soil surveys and in local field-scale

studies. However, unlike for conventional methods, the prediction accuracy of vis–NIR spectral models cannot yet be estimated before the data collection, which hampers its application at the local scale where often a high precision is required (e.g., field experiments). In this study we used soil data from six agricultural fields in Eastern Switzerland and calibrated i) field-specific (local) models and ii) general models (combining all fields) for organic carbon, total carbon, total nitrogen, permanganate oxidizable carbon and pH using partial least squares regression. 24 out of 30 local models showed an accurate or even excellent

performance (ratio of performance to deviation (RPD) > 2) and the root mean square errors (RMSE) of prediction were, except for pH, maximum five times higher than the lab measurement error. The variability of a specific soil property and the mean carbonate concentration in the dataset were the two factors influencing the performance of the local models. We found a significant relationship between the coefficient of variation in the dataset and the metrics for model performance ($R^2$, percental RMSE and RPD). Starting from a tolerable prediction error for the spectral measurements, the regressions can be used to

develop a sampling design that matches the corresponding target variability. The five inaccurately performing local models with RPD < 2 were on the two fields with highest carbonate content raising the question if local vis–NIR models are suitable for soils with high carbonate concentration. General models combining the datasets from all six fields showed an accurate overall performance but the RMSE on the field level were higher compared to the local models.

## 1 Introduction

The application of spectroscopy in the visible and near infrared (vis–NIR) range is increasing in soil science and related disciplines with the main objective to gain information on soil properties for more samples at lower costs than with conventional laboratory methods. Despite its often lower performance compared to mid infrared (MIR) spectroscopy, vis–NIR spectroscopy is widely applied because of less sample preparation, lower costs, and generally easier portability (Soriano-Disla et al., 2014). On-site measurements are therefore feasible but laboratory measurements with dried and sieved soil samples have

so far shown higher accuracy (Allory et al., 2019; Hutengs et al., 2019). In particular, soil properties related to soil organic





matter can be estimated appropriately by laboratory vis–NIR spectroscopy (Angelopoulou et al., 2020). In most cases, the focus is to provide soil information over large areas (e. g. soil maps) where a high sample number is present and only a moderate prediction accuracy is needed. Large-scale spectral libraries have been developed, to further reduce the need for wet chemistry data. Due to the high complexity within spectral libraries, the application of a general model to the local context

leads to high prediction errors. Recent research has shown that the localization of these infrared models substantially improves the predictive performance in a local context, for example by spiking (Brown, 2007; Li et al., 2020; Ng et al., 2022; Seidel et al., 2019; Wetterlind and Stenberg, 2010; Zhao et al., 2021), memory-based learning (Ramirez-Lopez et al., 2013), resampling algorithms (Lobsey et al., 2017) or deep learning (Shen et al., 2022). However, for analyzing small-scale variability (field or farm level), a local model is often still the best choice because it achieves the lowest prediction errors. On a theoretical level,

the development of local models is supported by the finding that in the vis–NIR range, spectral features that influence specific soil properties vary strongly between different datasets, which makes highly heterogenous large datasets prone to insufficient model performance (Angelopoulou et al., 2020; Grunwald et al., 2018).

The main motivation for the application of vis–NIR spectroscopy in local projects are lower costs for analytics that permits to increase the number of samples at only little additional costs. With a larger sample size, the spatial or temporal resolution can

be increased which allows conclusions about the within-field or within-farm variability but might potentially also increase the statistical power in agricultural experiments (Greenberg et al., 2022). The development of local spectral models has the main purpose to cope with a high sample size at the local scale, but these local models do not have any use beyond the analysis of the specific local dataset.

Spectral vis–NIR models developed from local datasets showed a very high variability in model performance ranging from

excellent models (Breure et al., 2022; Seidel et al., 2019) to those with relatively poor model performance (Camargo et al., 2022; Kuang and Mouazen, 2011). The reasons for these different performances of local models are understudied and remain often unclear. Among many different possible modelling approaches including support vector machine regression, artificial neural networks, cubist and random forest, partial least square regression (PLSR) is the most frequently used model type to build spectral models with small datasets (Alomar et al., 2021; Zhao et al., 2021).

Comparing the performance of several models is demanding because spectral models (as most other models) need several parameters for their evaluation and one parameter alone might be misleading. The most known statistical parameter is the goodness of fit ($R^2$) between the predicted versus the measured data. However, $R^2$ does not allow any conclusion about prediction accuracy. The root mean square error (RMSE) is the estimated prediction error of the spectral model and has the unit of the measured property. However, it is difficult to compare the RMSE among models and datasets since the data range

varies. Therefore, the ratio of performance to deviation (RPD), which relates the RMSE with the range of the data expressed as standard deviation, is often used. A less frequently reported but important performance parameter is the bias that expresses the average over- or underestimation of the model predictions. The bias is particularly important if the model is applied on samples that are very different from the samples used in the calibration model (Bellon-Maurel and McBratney, 2011).



For local models at field or farm scale, the variability of the target soil property is very important (Alomar et al., 2021; Perez-Bejarano and Guerrero, 2018; Stenberg et al., 2010). If the variability is very small, spectral measurements might not be sensitive enough to distinguish the different samples, which may result in a low RMSE but also in a low $R^2$ and probably also a low RPD. On the other hand, if variability of a specific soil property is high the local spectral models tend to have a higher $R^2$ and eventually higher RPD, but also a higher RMSE as observed by Kuang and Mouazen (2011). However, it remains unclear what an optimal variability of a soil property in a project of local extent would be to achieve a high measurement accuracy in absolute values (low RMSE) but also relative to the range of the data (RPD).

Besides dataset variability, the number of samples is crucial for local models because, often, only a limited number of samples with reference laboratory data are available. It has been shown that local models improve with increasing numbers of calibration samples and that a sample size of at least 50 provides accurate prediction models (Kuang and Mouazen, 2012). Some studies thus combined multiple target sites and develop a general model by combining all the local datasets to reach a larger sample size and potentially better model performance (Kuang and Mouazen, 2011; Singh et al., 2022). In these studies, the general model showed an intermediate performance, and the general prediction error was between the best and the poorest performing local model. However, these studies only calculated the overall prediction error of the general model and therefore it is not clear if the prediction on target sites with poorly performing local models could be improved by applying a general model.

For vis–NIR spectroscopy application at local scales, it is therefore very difficult to estimate the measurement accuracy for the predicted samples beforehand. This uncertainty is probably the main reason that hampers the application of vis–NIR spectroscopy because researchers prefer to rely on conventional lab measurements with a smaller sample size (and smaller spatial resolution) where the measurement accuracy is known before sampling and measurements are conducted. Applying spectroscopy at field or farm scale thus bears the risk that the measurement accuracy (RMSE) may be beyond the tolerable threshold, which might then question the whole project. Thus, in this paper, we analyze the performance of field-specific (local) spectral models of a field experiment conducted in six fields in Eastern Switzerland and that of a general model combining the data from all six fields to ascertain their influencing factors. We ask:

1.     To what extent do the prediction errors of local spectral models differ from the lab measurement error?

2.     Does a general model that includes several target sites improve the prediction on a target site with a poor local model performance?

3.     What is the optimal variability in the dataset of a local model to achieve a good model performance?

4.     Which field and soil characteristics (field size, soil texture, carbonate concentration) of the target site influence the performance of spectral models?

Answering these questions, we want to provide insights for estimations of prediction accuracies for vis–NIR studies at the local scale with the objective to support decision making during the development of a sampling design.



## 2 Methods

### 2.1 Datasets from a cover cropping experiment on six field sites

We used datasets from six fields (A, B, C, D, E, F) of a cover cropping experiment in the Canton of Thurgau, Eastern Switzerland (paper in preparation). The six fields were maximally 13 km apart from one another and soil type was for all of them Eutric Cambisol that had developed on base moraine (Table 1). The aim of the study was to compare the influence of two different cover cropping regimes on short-term soil organic matter cycling. Each field had 39 differential GPS (dGPS) referenced sampling points in an unaligned sampling design. At each dGPS referenced point, soil was sampled three to four times in three depths (0-5, 5-10 and 10-20 cm) during one long cover cropping period (August 2019 to May 2020). Fields A, B, C and D had four sampling times resulting in 468 samples per field. Fields E and F had three sampling times resulting in 351 samples per field. All samples were dried at 40° C to constant weight (around 72 h) and then gently crushed and sieved to 2 mm. For the total sample size of 2574 samples, soil properties were estimated using vis–NIR soil spectroscopy, whereas 386 samples were analyzed conventionally by wet chemistry for subsequent calibration modeling. These 386 samples for laboratory analysis were selected for each field separately using the Kennard-Stones algorithm (Kennard and Stone, 1969) to ensure a cover of the whole spectral variability. Thereby, the Kennard-Stones algorithm was run with 2 to 7 principal components and the number of principal components was chosen that covered at least 99 % of the spectral variance and provided a reference sample selection that well represented the different sampling times, soil depths and spatial distribution. The laboratory analysis comprised soil organic C (SOC), total C, total N, permanganate oxidizable C (POXC) also referred as active C, and pH.

### 2.2 Chemical soil analyses and its accuracy

Total C and N concentrations were measured on a ground aliquot by dry combustion (vario MICRO tube, Elementar, Germany). Inorganic C was analyzed for each sample in triplicates through the dissolution of carbonate in a Scheibler-apparatus with 10 % HCl solution and the measurement of the evolved CO2 volume. SOC was then calculated as the difference between total C and the mean of the three measurements for inorganic C. POXC was measured according to the Protocol of Weil et al. (2003) with the adaption of Lucas and Weil (2012). In brief, 2.0 mL of 0.2 M $KMnO_4$ were added to 2.5 g of soil and after a reaction time of totally 10 minutes, the absorption of the liquid was measured at 550 nm with a Spectrophotometer (UV-1800, Shimadzu corporation, Japan). Measurement of pH was done in a 0.01 M $CaCl_2$ solution. Mean soil texture was measured for each field on a representative composite sample with the improved integral suspension pressure method (ISP+; Durner and Iden (2021)) on a PARIO Plus Soil Particle Analyzer (METER Group, Germany/USA).

To estimate the lab measurement error, we took three samples per field (in total 18) where we conducted the measurements for total C, total N, POXC, and pH in triplicates to calculate a standard deviation. To estimate the measurement error of SOC we took the sum of the standard deviation of the total C measurement with the standard error of the inorganic C measurement because inorganic C measurement were for all samples done in triplicates. The measurement error of all 18 triplicates were then averaged to obtain the overall lab measurement error for a soil property.



## 2.3 Spectral measurement and pre-processing of spectra

All samples were measured with a vis–NIR spectrometer (ASD FieldSpec 4 Hi-Res, Malvern Panalytical, USA) with a
sampling interval of 1.4 nm from 350-1000 nm and 1.1 nm from 1000-2500 nm. The device then provides a reflectance
spectrum with a resolution of 1 nm and 2151 wavelengths. Measurements were done with a contact probe, containing an
internal halogen bulb, which was in a fixed position and soil samples, placed in a petri dish of 1.5 cm height and 3 cm diameter,
were lifted with a laboratory scissor jack until close contact with the probe to ensure a stable measurement position. For each
sample, five petri dishes were filled to provide five replicate spectra per sample. Each of these five replicates consists of 30
internal repetitive scans that were automatically averaged by the device internal RS3 software. Between samples, the contact
probe was carefully cleaned with water and ethanol. After the 5 replicates of a sample, the calibration of the spectrometer was
checked with a 100 % reflectance white reference panel (Spectralon, 12×12 cm, Labsphere, USA). The infrared data of each
sample was kept in two versions, once as reflectance spectra, as provided by the spectrometer, and once as absorbance spectra
using the log(1/reflectance) transformation. Several pre-processing options and their combination were tested on both, the
reflectance and the absorbance spectra: a) resampling of the spectra in an interval from 1 to 6 nm, b) cutting of the beginning
(350-400 nm) or the end (2450-2500 nm) of the spectra c) first or second order derivative d) Savitzky-Golay (SG) smoothing
in a polynomial order of 3 with window sizes ranging from 5 to 51, e) gap segment derivative (GSD) with window width
between 5 and 51 and segment size between 1 and 21, f) standard normal variate (SNV) combined with GSD, g) SG smoothing
combined with multiplicative scatter correction (MSC). All applied pre-processing techniques are frequently used in soil
spectroscopy and well described in Ellinger et al. (2019). The pre-processing techniques from a) to g) led to around 100
meaningful combinations that were tested in model building and the final pre-processing option was selected based on the
lowest RMSE.

## 2.4 Development and evaluation of field-specific local models

We used for all 30 local models (6 fields x 5 properties) a PLSR modelling approach (Wold et al., 1983). Since all soil
properties showed a limited skewness (see Table S1 in the supplementary material) that was always in the range of $-2 \leq skew \leq 2$ which was proposed as acceptable to assume a normal univariate distribution (George and Mallery, 2010), we consider the
application of PLSR appropriate, especially since it is robust to minor deviations from a normal distribution (Goodhue et al.,
2012).

Model performance was assessed using the statistics of the hold-out folds of each five times repeated five-fold cross-validation
because it was evaluated as a robust method for smaller datasets (Kuhn and Johnson, 2013; Molinaro et al., 2005). To avoid
model overfitting, we set the maximum of latent variables in the PLSR model to 12. For each number of latent variables (1, 2,
…., 12) the dataset was five times randomly split into five folds of which four were used for model training and the remaining
fold was held out and used for model validation. The RMSE (Eq. 1) of the hold-out samples was averaged among the five
repeats resulting in a cross-validated RMSE per number of latent variables. The final number of latent variables was then



chosen according to the "one standard error rule" which means that instead of directly choosing the number of latent variables
with lowest mean RMSE, the most parsimonious (less latent variables) model within one standard error of the mean RMSE of
the optimal model was selected (Hastie et al., 2017). The final model was trained using all training data with optimized number
of latent variables.

A proper validation of a spectral model is very crucial and particularly important in this study where soil was repeatedly

sampled in different depths at the same GPS point. To analyze the correlation among the samples and define a grouping factor
for the cross-validation, we calculated the mean Euclidean distances between all samples and compared it with the mean
distance 1) between samples at the same GPS point but different depths, 2) between samples at the same point and depth but
different sampling times and 3) between samples at the same point but different depth and sampling times (Fig. S1 in the
supplementary material). Thereby, we have seen that the soil samples from the three different soil depths sampled at the same

GPS-point at the same sampling time had a substantially lower mean Euclidean distance compared to the overall mean.
Consequently, we grouped the samples from the same GPS point at the same sampling time and kept them in the same fold to
avoid a too optimist model evaluation during cross-validation.

Since we used a cross-validation approach on the field scale, all models showed a very small bias. We therefore do not discuss

the bias in this paper and focus on $R^2$, RMSE and RPD (Eq. 2) for the evaluation and comparison of different models. RMSE
was calculated according to Equation 1 where $\hat{y}_i$ is the prediction of the spectral model and $y_i$ the actual measured value in the
laboratory.

$$RMSE = \sqrt{\frac{1}{n}\sum_{i=1}^{n}(y_i - \hat{y}_i)^2} \, ,$$    (1)

RPD compares the RMSE with the standard deviation (SD, Eq. 2) of the data

$$RPD = \frac{SD}{RMSE} \, ,$$    (2)

To classify the model performance, we combined the RPD based classification of Chang et al. (2001) and Zhang et al. (2018).
We considered spectral models with RPD < 1.4 as poor, models with RPD between 1.4 and 2 as approximate, models with
RPD between 2 and 3 as accurate and models with RPD > 3 as excellent. Even though in spectroscopy project of a local extent
the RMSE is probably the most important model performance parameter, RPD is the best parameter to compare models of

different scales. Model metrics ($R^2$, RMSE and RPD) mentioned in the text are based on the cross-validation and metrics for
the model calibration in Table 2 are specifically labelled as $R^2_{cal}$, $RMSE_{cal}$, $RPD_{cal}$.

### 2.5 Development and evaluation of general models

In addition to the field-specific local models, we built general models for the five soil properties that included all reference
samples (n = 386) of the six fields. Even though for this sample size an independent test-set would be more suitable than a





cross-validation approach, we evaluated the model performance using the hold-out samples in the five times repeated 10-fold cross-validation, keeping, as for the local models, samples from same GPS point and same sampling time in the same fold. The first reason for not using an independent validation set is that the modelling approach of the general model should be similar to the one of the local models to make them comparable. The second reason is that a representative split of the dataset into a calibration and a validation set according to the spectral variability would not result in equal number of samples per field

in the validation set. Conversely, if we selected an equal sample size per field for the validation set, we would not have been able to cover the entire spectral variability. Evaluating the general models with hold-out samples of the cross-validation allowed us to calculate the RMSE over all samples but also the RMSE for the samples of each field individually and compare it with the RMSE achieved by the local models. Since the only purpose of the general models was to increase modelling efficiency for a specific combined dataset, we did not group the samples according to fields during cross-validation because

the same share of samples from the same field would also be in the prediction dataset.

**2.6 Model interpretation**

To interpret spectral models, it is crucial to find relevant spectral features that are consistently important for a certain soil property. To identify the most important wavelength ranges in the final chosen models we used the variable importance in projection (VIP) method first published by Wold et al. (1993) and evaluated by Chong and Jun (2005). The VIP method can

deal with multicollinearity and is therefore suitable for the interpretation of spectral models (Baumann et al., 2021). Wavelengths that have an above-average impact on the model have a VIP score above 1. We classified spectral ranges in groups of VIP scores between 1 and 1.5, 1.5 and 2 as well as VIP scores above 2.

**2.7 Assessment of site characteristics influencing model performance**

The 35 developed spectral models differed in their performance. Therefore we tested various factors that might explain under

what field and soil characteristics the application of vis–NIR spectroscopy is most beneficial at the local scale. We focused on the relationship between variability of a soil property in a dataset and model performance. We looked at the coefficient of variation (CV), also known as the relative standard deviation, of the modeled soil property and examined its relationship with $R^2$, RMSE and RPD. Thereby, we expressed RMSE in percentages of the mean (PRMSE) to make all models comparable. We used linear regressions to fit the PRMSE and the RPD with the CV. Since $R^2$ cannot be higher than one, we fitted a two-

parameter Weibull function (Seber and Wild, 2004) to the CV. For each regression we calculated the 95 % confidence and the 95 % prediction interval. In the discussion part, we compare the significant regressions between CV and model metrics found in our datasets with model performance in literature. Since the 95 % prediction interval for RPD independence of CV was quite high, we only discuss the found regressions for $R^2$ and PRMSE with the performance of local models in the literature. Furthermore, we fitted a linear regression for RPD in the range $0 < CV < 44$ % but we expect that with higher CV, RPD might

also show saturation and therefore an extrapolation of our fitted linear regression would not make sense. In addition to the





variability in the dataset, we analyzed the influence of mean carbonate concentrations, soil texture and field size on the model metrics but only the mean carbonate concentration showed effects and is therefore presented in the results.

**2.8 Data organization**

All analysis were performed in R version 4.0.3 (R Core Team, 2020). The spectral datasets were analyzed using the R-package
simplerspec (Baumann, 2019) in combination with the packages prospectr (Stevens and Ramirez-Lopez, 2020) and caret (Kuhn, 2020).

**3 Results**

**3.1 Description of the datasets**

A comparison of the data distribution between the six different fields can be seen in Fig. 1 and the corresponding statistics in
Table S1 in the supplementary material. The means for SOC, Total N and POXC differed between the six fields, but the distribution was relatively similar for these three soil properties. The density functions for total C and pH were highly influenced by the spatial distribution of carbonate in the soil: Fields B, D and E contain samples with and without carbonate resulting in a broad distribution for both, total C, and pH. All soil samples of fields A and C contained carbonate in varying concentrations resulting in a broad distribution for total C but narrow distribution for pH. Field F showed high and only slightly
varying carbonate concentrations and therefore a very narrow distribution for total C and pH.

**3.2 Performance of spectral models**

Based on RPD, 13 out of 30 local models showed an excellent performance (RPD > 3), 12 models an accurate performance (RPD > 2), four models an approximate (RPD > 1.4) and one model a poor performance (RPD > 1.4; Table 2). The five models without accurate performance were from field A (SOC, POXC and pH) and field F (SOC and pH).
However, the RMSE of the local models for pH of fields A (0.07) and F (0.03) were lower than the RMSE of the other three local models (between 0.13 and 0.18) whose performance was classified as accurate. Differently, the local models for SOC on fields A and F with only approximate performance showed a higher RMSE (1.88 and 2.42 g kg$^{-1}$) than the other accurately performing local models for SOC (between 1.05 and 1.58 g kg$^{-1}$). The five general models showed all an accurate performance with RPD ranging from 2.42 to 3.97.

**3.3 Influence of pre-processing on spectral variability**

There was no pre-processing combination that proved to be suitable for all models of the same field (Fig. S2 in the supplementary material) or of the same soil property (data not shown). Nevertheless, for all 35 models, pre-processing was necessary and models could be improved compared to the raw spectra. Most times, the first or second order derivatives



improved the models substantially. Most models performed best when the spectra were reduced to every third wavelength and
models based on absorbance were a bit more frequently used than models based on reflectance. The combined application SG
filter and MSC was the most successful pre-processing while a single SG filter, GSD and SNV in combination with GSD were
of minor importance. Cutting of the beginning (350-400) or end of the spectra (2450-2500) sometimes improved the model
performance but since most pre-processing steps reduce the beginning and end of the spectra, it was not possible to evaluate
the cutting. Similarly, it was not possible to evaluate the window width chosen in the SG-filter because there is an interference
with the resampling interval. A detailed list of the selected pre-processing options of the final models and the corresponding
metrics for model performance can be found in Table 2.

The sensitivity of model performance to pre-processing can be visualized with the biplots of principal component analysis
(PCA). Figure 2 shows the first three biplots of the raw spectra and the spectra that were pre-processed according to the general
models of the five soil properties. The raw spectra had a very high share of the explained variance (96.8%) on the first principal
component but hardly any groups according to fields could be observed with the first two principal components. All pre-
processing options used for the general models decreased the explained variance on the first principal component (32.5 to 39.6
%) and a grouping according to fields could already be seen in the biplot of the first two principal components. Thereby,
especially field F with highest carbonate concentration and field C with highest clay content often showed clear groups.
Nevertheless, in the pre-processing for pH, field E with the highest pH variability shows a clear group in the first biplot and
the pH variability is well represented with the first PC.

## 3.4 Comparison of general models with local models and lab measurement error

The overall cross-validated model metrics of the general model (black filled circle in Fig. 3) indicated over all fields for all
soil properties a good performance, but the field-specific model evaluation showed distinct differences among fields. The field-
specifically calculated $R^2$ of the general models of fields B, C, D and E was similar to the $R^2$ of the local model for SOC, total
C, total N and POXC (only a slight slope in Fig. 3). For pH only field C and D showed similar $R^2$ in the local and general
model while all other fields showed clearly higher $R^2$ in the local model. On the other hand, field F had clearly lower $R^2$ in the
general model than in the local model (strong negative slope) for all five soil properties except POXC. For field A, $R^2$ was
similar between the local and the general model for total C but for the other four properties clearly lower in the general model.
The field-specifically calculated RPD of the general model was across all soil properties on average 27 % lower compared to
the local models (Fig. 3). All property-field combinations of fields B, C, D and F showed at least an approximate (RPD > 1.4)
performance in the general models, whereas the eight poorly (RPD < 1.4) performing property field combinations were all
from fields A and F. It can therefore clearly be concluded that the general models could not improve the low performing local
models.

Field-specifically calculated RMSE of the general models was on average 49 % higher compared to the local models. However,
there were substantial differences between the different fields. Field A and F showed the highest increases of field-specific



RMSE in the general model compared to the local model for SOC, total C, and total N (strong slopes in Fig. 3). On the other hand, field E showed quite similar RMSE in the local and in the general model for all soil properties except total C.

The RMSE of the best local models was close to the overall lab measurement error for SOC, total C and total N, a bit higher for POXC and substantially higher for pH (Fig. 3). The overall lab measurement error for SOC (1.3 g kg$^{-1}$) was calculated as

the sum of the measurement error for total C and inorganic C and therefore the RMSE for SOC on fields with only little inorganic C (fields B and D) was lower than the overall lab measurement error. However, the RMSE of SOC on field B (1.20 g kg$^{-1}$) and D (1.05 g kg$^{-1}$) with little inorganic C was clearly above the lab measurement error of total C which would for these two fields be the better lab reference error (0.85 g kg$^{-1}$). The cross-validated RMSE of the local spectral models exceeded the overall lab measurement errors between factor 0.8 and 1.9 for SOC, 1.4 and 3.0 for total C, 1.3 and 2.0 for total N, 2.1 and 4.0

for POXC and between 2.8 and 16.5 for pH. The field-specifically calculated RMSE of the general model exceeded the overall lab measurement error between factor 1.0 and 2.1 for SOC, 2.4 and 4.9 for total C, 1.5 and 3.3 for total N, 2.7 and 4.9 for POXC and between 7.9 and 22.4 for pH.

The VIP scores (Fig. 4) show that the most important wavelengths were very dataset-specific and differed for the same property substantially from one another. However, for all the analyzed soil properties the wavelength ranges between 400 and 750 nm

(visible) as well 1800 and 2450 nm were most important while the range in between was of lower importance. Nevertheless, some models had VIP scores above 2 in the range between 750 and 1800 nm. Prediction performance in terms of RMSE and RPD of total C of fields E and F was particularly lower in the general model than in the local model (Fig. 3). This finding can be explained with the VIP analysis (Fig. 4) that showed for the general model the most important wavelength range between 2150 and 2450 nm but for the local models of fields E and F in the range of 500 to 1020 nm. The local model for total N of

field F showed very high VIP scores (>2) in a small specific range between 2345 and 2369 nm but these wavelengths were not important in the general model for total N (Fig. 4), which resulted in a much lower prediction accuracy of total N for field F in the general model compared to the local model.

### 3.5 Influence of dataset variability on model performance

The three model performance parameters, $R^2$, RPD and PRMSE increased with increasing CV in the dataset (Fig. 5). For the

relationship of $R^2$ and CV both parameters of the Weibull function were significant (p < 0.05), and the function shows that above a CV of 10 % the $R^2$ of the spectral models only show minor increases with increasing CV. The PRMSE showed a relatively strong linear relationship with CV ($R^2 = 0.61$, p < 0.001) and the regression slope was higher than the slope of the regression for RPD which showed a weaker relationship with CV ($R^2 = 0.40$, p < 0.01). The reason is that RPD accounts for the variability in the dataset because by its definition the increasing RMSE is divided by an increasing standard deviation as

the CV increases, which makes RPD less influenced by the CV in the dataset. Nevertheless, we found a significant increase of RPD with increasing CV indicating a better performance of spectral models with higher variability in the dataset at least up to a CV of 44 %. The optimal variability in a spectroscopy project of local extent might mainly be limited by the PRMSE that increases relatively fast and consistent with increasing CV in the dataset. The 95 % prediction intervals in Fig. 5 show that a





dataset with CV of 15 % is sufficient to reach an $R^2$ in the spectral models above 0.65 and keeping the PRMSE at the same time between 1.8 and 8.8 %. For RPD the 95 % prediction interval for a new dataset with 15 % CV is very wide ranging between 1.2 and 4.4. An increase of CV in the dataset provides slight increases of $R^2$ and RPD but PRMSE increases faster and at a CV of 40 % the PRMSE would range between 6.9 and 14.5 % with 95 % prediction probability (Fig. 5). For many projects this might still be tolerable for others it might already be too high.

## 3.6 Site characteristics influencing model performance

Besides the variability for a specific property in the dataset, the performance of the local field-specific models for SOC were influenced by the mean inorganic C concentration. The local models of fields A and F with highest carbonate concentrations showed, aggregated over all five soil properties, lower $R^2$ and RPD than the other fields (Fig. 6).

## 4 Discussion

## 4.1 Performance of local spectral models

Most of the developed local models showed an accurate performance and confirm the suitability of vis–NIR spectroscopy in projects of local or single plot extent. The performance (based on RPD) of the two models for pH on field A and F, which were classified as only approximate or even poor, respectively, can be explained by the low variability of pH in these datasets (see Fig. 1) and is supported by the fact that these two models have the lowest RMSE values for pH (Fig. 3). This explanation does not hold for the other three local models that were also classified as only approximate because SOC and POXC on field

A as well as SOC on field F showed a similar variability as on the other fields (Fig. 1). We assume that the lower performance of these three models on field A and F is caused by the parent material containing a high carbonate concentration (see section 4.6). Nevertheless, in all local models the RMSE was mostly comparable to the lab measurement error which was also the case in the field scale models with MIR spectroscopy of Greenberg et al. (2022). In agreement with literature (Soriano-Disla et al., 2014), primary properties with a direct impact in the vis–NIR range like SOC, total C, total N, and POXC showed a RMSE

that was closer to the lab measurement error (maximum four times higher). On the other hand, pH has only an indirect impact on the spectra and thus showed a much higher RMSE compared to the lab measurement error (maximum 16.4 times higher).

## 4.2 Comparison of general models with local models

The general models could not improve the prediction of low performing local models. This finding is especially interesting because in this study the general model was built with datasets of six fields that were spatially close to one another (maximal

distance of 13 km), had the same soil type and the same parent material. However, the base moraine as a parent material can be variable which we mainly observed in different soil texture and carbonate concentration but also in the high spectral variability (see PCA biplots in Fig. 2). In this sense we confirm the conclusions of Seidel et al. (2019) and Ng et al. (2022) who suggested that the best solution is always to develop a local model if enough samples (> 30) are available. This conclusion





is supported in this study by the quite distinctive pattern of VIP scores between the different models (Fig. 4). The overall

picture shows that the wavelengths between 2000 and 2450 nm followed by the visible range between 400 and 700 nm were most important for prediction of the investigated properties, which is in agreement with the literature (Munnaf and Mouazen, 2022; Soriano-Disla et al., 2014). Nevertheless, each local model has distinct and site-specific features that could not be attributed to specific soil characteristics but obviously were important for the model development. The development of general models where different locations are aggregated in one dataset can save costs because the number of lab analysis per location

can be reduced and less work is required for model building. Depending on the research purpose and the required measurement accuracy, the development of general models can be a very suitable and cost-effective approach. Nevertheless, this study showed that some fields (A and F) can show a poor performance in general models, hence it is crucial to consider what locations or datasets are being combined.

## 4.3 Pre-processing

The selection of the optimal pre-processing scheme was crucial for model performance but strongly dependent on the dataset. Often MSC was the best performing pre-processing option, which was confirmed in some studies (Cambule et al., 2012; Liu et al., 2019) but disproved in others (Knox et al., 2015; Riefolo et al., 2020). We therefore highly recommend considering MSC as a pre-processing option in spectral modelling but at the same time agree with Barra et al. (2021) that there is no general pre-processing solution that works for all datasets. The principal component analysis with the combined dataset of all fields

(Fig. 2) illustrates this finding by the different grouping of individual field datasets due to different pre-processing. This leads to the conclusion that studies that did not optimize the pre-processing scheme for every soil property separately did eventually not make full use of the spectroscopy potential (see also examples in Table 3), which has been shown by other studies as well (Alomar et al., 2021; Rodriguez-Febereiro et al., 2022; Singh et al., 2022). Nevertheless, the property-specific optimization of spectral pre-processing is a tedious process and constrains the fast and cost-effective application of vis–NIR spectroscopy, but

some progress has been made simultaneously with our study by Mishra et al. (2022).

## 4.4 Influence of dataset variability on model performance

With 35 spectral models we found significant relationships between CV in the dataset and model performance. Hereby it must be stated that the built regressions only have a validity for local datasets with relatively similar mineralogy. A dataset could potentially also have a very low CV for a certain soil property but could contain soil samples from very distinct regions with

very different mineralogical background. Nevertheless, we also included the general models in that regression knowing that the different fields differ in some respects but are not fundamentally different from one another since they are in the same region. The combined dataset used for the general model contains a higher variation of soil samples but does not necessarily have a higher CV for a certain soil property. Nevertheless, it can be seen in Fig. 5 that the general models (black symbols) fit quite well into the regression for $R^2$, PRMSE and RPD. Dataset variability has always been an issue in vis–NIR spectroscopy

application and many authors stated that "enough" variability is necessary (Alomar et al., 2021; Perez-Bejarano and Guerrero,



2018) without directly quantifying it. Based on the 95 % prediction interval of the found regression we suggest a minimum CV of about 15 % for a successful spectral model. The question of an optimal maximum variability in a local dataset for spectral modelling has so far not really been addressed. In many studies, a better model performance is ascribed to a higher variability in a dataset (Kawamura et al., 2021; Nawar and Mouazen, 2019) without considering the increasing prediction error.

The observation that a higher dataset variability leads to a higher RMSE in local spectral model was also made by Kuang and Mouazen (2011) but so far, no quantitative relationship was provided. Our results show clearly that a higher variability in a dataset linearly increases the prediction error (PRMSE). Therefore a dataset for a local vis–NIR spectral model has an optimal maximum variability that depends on the tolerable prediction error in a research project.

### 4.5 Comparison of found regressions with performance of local models in literature

The comparison of our fitted regressions for $R^2$ and PRMSE with performance of local models in literature can be found in Fig. 7 where each number of a data point corresponds to a publication listed in Table 3. Thereby, we only plotted local models with a CV < 105 %. The $R^2$ found in literature lie often within the prediction interval for CV > 25 % but for lower CV our nonlinear regression seems to be too optimistic (Fig. 7). For the PRMSE, our linear regression result tends to be too optimistic over the whole range of CV but especially for datasets with CV > 60 %. The main reason for this difference in performance

may be that we developed field-specific models covering an area of about 1 ha while most local models found in literature cover larger areas, which might not necessarily result in a higher CV for a certain soil property but in tendency increases the variability in the mineralogical background. However, the models that were performing the furthest outside of the 95 % prediction interval for $R^2$ belong all to study 19, followed by models from study 3, 2 and 20. For PRMSE, the models showing a far higher PRMSE than the upper 95 % prediction interval belong to studies 4, 19, 1, 7, 3 and 13. It can be seen that in these

studies showing a relatively poor model performance, only study 1 and 7 optimized pre-processing for each soil property individually. Additionally, for properties with a low mean like in study 4 (mean SOC = 0.79 %), the high PRMSE is misleading because the absolute error was relatively low. The suggested regression for PRMSE can serve as a rough estimate for model performance but with a higher CV the PRMSE shows higher variability. A vivid example is study 7 (Singh et al., 2022), with four field-specific models and one general model for SOC, that shows $R^2$ values that lie all within the 95 % prediction interval,

but the PRMSE values vary around the suggested 95 % prediction interval. In this respect, we consider the 95 % prediction interval for $R^2$ as a good predictor for the expected $R^2$ of local spectral models and recommend a minimum CV of 15 % in a dataset for local models. The regression for PRMSE gives a rough estimate about the error that must be expected in a spectroscopy project of a local extent, but the variability is relatively high because it is strongly influenced by the overall mean in the dataset and other factors that remain unknown and are probably related to the mineralogical background.

### 405 4.6 Site characteristics influencing model performance

We found an influence of carbonate concentration with lowest performance of local spectral models on fields A and F. Similar observations were made by Amare et al., (2013) and McCarty et al. (2002) who argued that the absorbance bands of carbonate





mask those of SOC. In this context, Reeves (2010), who showed that the spectrum of a soil sample varied greatly with its carbonate concentration, considered the prediction of SOC in soils with high carbonate concentration as one of the open

questions in vis–NIR spectroscopy research. Even though the local models of fields A and F had the lowest performance among all local models, they still showed, with exception of pH, approximate results which, depending on the research question might still provide useful information. An important point missing in this discussion is the measurement accuracy of SOC in the laboratory which is strongly influenced by the presence of carbonate and the method used (Goidts et al., 2009). If the soil samples contain carbonate, often two measurements must be conducted, and SOC is calculated as the difference between total

C and inorganic C. Especially with a high carbonate concentration the measurement error for the inorganic C concentration can be a substantial share of the SOC concentration. The higher lab measurement error with higher carbonate concentration can explain the lower model performance on soils with high carbonate concentration for SOC but not for the other four soil properties where model performance still tended to be lower than on fields with little carbonate concentration (Fig. 6). This confirms the above-mentioned observation of spectral interference between inorganic C and organic matter and is additionally

substantiated by the result that most properties of fields A and F showed a poor performance in the general models (Fig. 3). It is known that carbonate has many more defined peaks and less interferences with organic matter in the MIR than in the vis–NIR (Reeves, 2010), therefore datasets that combine soil samples with high and low carbonate concentrations might better be predicted with MIR spectroscopy. Unlike Stenberg et al. (2010) and Heinze et al. (2013), we did not find a better model performance with increasing mean clay content in the dataset which might also be explained by the relatively small range in

mean clay contents between 18 % (Field F) and 38 % (Field C).

## 5 Conclusion

This study investigated the impact of various factors on vis–NIR modeling performances and compared a local and a general modeling approach. Among the 35 built, models 30 performed accurate or even excellent whereby the RMSE was maximally four times higher than the lab measurement error for all analyzed soil properties except pH. We identified the variability and

the mean carbonate content in a dataset as crucial factors that influence the performance of local models. We found significant relationships between the CV in a dataset and the model performance in terms of $R^2$ and PRMSE. The lower 95 % prediction interval indicates that for a good $R^2$ ($> 0.65$) a CV of at minimum 15 % is required in a dataset. Assuming a tolerable prediction error of 12 %, the upper 95 % prediction interval for PRMSE would suggest a maximum CV of 30 % in the dataset. Since from our experience the main concern to use vis–NIR spectroscopy in a project of local extent is the unknown measurement

accuracy, we recommend determining first a tolerable percental measurement error and then developing a sampling design according to the targeted variability in the dataset. Such a procedure requires an attentive stratification strategy for soil sampling according to land use, management, soil depth or parent material. Since the variability of different soil properties can be very different, as e.g., on cropland SOC normally shows much higher variability than pH, we recommend to clearly



prioritize the soil properties and select the one of most interest and optimize the sampling design according to that target
property. Fields with high carbonate content (A and F) showed lower (but often still approximate) performance in the local
models but a strongly decreased performance in the general models compared to fields with low carbonate concentration. We
therefore conclude that in case of low carbonate concentration, several target sites of a region can be combined to a general
model with only a minor reduction in model performance. Whether or not several target sites with high carbonate concentration
can be combined in one general model using vis–NIR spectroscopy is a question that is not answered yet and requires further
research. However, since carbonates show less interferences with organic matter in the MIR than in the vis–NIR spectral range,
soil samples from sites with high carbonate concentration might be the better predicted with MIR spectroscopy. Considering
the mentioned recommendation, the application of laboratory vis–NIR spectroscopy in projects of local extent provides the
opportunity to increase, at low costs, the sample size drastically and increase, as desired, the spatial or temporal resolution in
the sampling design with only minor decreases in measurement accuracy.

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





**Table 1: Description of the datasets of the six different fields A to F. All fields were classified as Eutric Cambisol developed on base moraine. Soil texture was measured with the improved integral suspension pressure method (ISP+)**

| Field | Coordinates | Elevation [m above sea level] | Area [ha] | Mean soil texture (Sand/Silt/Clay) [%] | Number of samples | |
|---|---|---|---|---|---|---|
| | | | | | Spectroscopy | Wet chemistry |
| A | 47° 40' 58" N / 08° 45' 54" E | 420 | 0.84 | Sandy loam (50/29/21) | 468 | 70 |
| B | 47° 40' 54" N / 08° 46' 05" E | 420 | 0.67 | Sandy loam (44/35/20) | 468 | 70 |
| C | 47° 38' 01" N / 08° 57' 02" E | 600 | 0.44 | Sandy loam (27/35/38) | 468 | 70 |
| D | 47° 38' 43" N / 08° 42' 58" E | 460 | 0.64 | Clay loam (28/44/28) | 468 | 70 |
| E | 47° 38' 49" N / 08° 43' 06" E | 460 | 1.05 | Sandy loam (30/48/23) | 351 | 53 |
| F | 47° 34' 22" N / 08° 48' 52" E | 380 | 0.3 | Sandy loam (39/43/18) | 351 | 53 |

**Table 2: Description of applied pre-processing and model performance of the final chosen models using a Partial Least Square regression. The local models (fields A to F) were evaluated with 5 times repeated 5-fold cross-validation and the general models (All) with 5 times repeated 10-fold cross-validation. RMSE = Root mean square error, RPD = ratio of performance to deviation, Refl. = Reflectance, Abs. = Absorbance, SG = Savitzky-Golay filter (m = order of derivative, w = window width), SNV = standard normal variate, GSD = gap segment derivative (m = derivative, w = window width, s = segment size), MSC = multiplicative scatter correction**

| Field | Property | Range of wavelengths / interval [nm] | Pre-processing | Latent variable | n | Calibration | | | Cross-validation | | | Model performance |
|---|---|---|---|---|---|---|---|---|---|---|---|---|
| | | | | | | $R^2_{cal}$ | $RMSE_{cal}$ | $RPD_{cal}$ | $R^2$ | RMSE | RPD | |
| A | SOC [g kg$^{-1}$] | 350-2500 / 3 | Abs., GSD (m=4, w=21, s=21) | 2 | 70 | 0.60 | 2.33 | 1.59 | 0.57 | 2.42 | 1.53 | Approximate |
| | POXC [g kg$^{-1}$] | 350-2500 / 2 | Refl., SNV, GSD (m=2, w=31, s=1) | 7 | 70 | 0.81 | 0.05 | 2.28 | 0.67 | 0.06 | 1.74 | Approximate |
| | Total N [g kg$^{-1}$] | 370-2480 / 6 | Refl., SG (m=1, w=21), MSC | 7 | 70 | 0.87 | 0.11 | 2.77 | 0.79 | 0.14 | 2.16 | Accurate |
| | Total C [g kg$^{-1}$] | 390-2500 / 4 | Abs., SNV, GSD (m=2, w=21, s=1) | 6 | 70 | 0.94 | 2.14 | 4.21 | 0.92 | 2.58 | 3.49 | Excellent |
| | pH | 410-2500 / 4 | Abs., SG (m = 1, w=35) | 5 | 70 | 0.74 | 0.06 | 1.97 | 0.65 | 0.07 | 1.69 | Approximate |
| B | SOC [g kg$^{-1}$] | 360-2500 / 5 | Abs., SG (m=2, w=21), MSC | 7 | 70 | 0.98 | 0.66 | 6.47 | 0.92 | 1.20 | 3.55 | Excellent |
| | POXC [g kg$^{-1}$] | 360-2480 / 3 | Abs., SG (m=2, w=21), MSC | 4 | 70 | 0.94 | 0.03 | 4.08 | 0.85 | 0.04 | 2.57 | Accurate |
| | Total N [g kg$^{-1}$] | 360-2480 / 5 | Abs., SG (m=2, w=21), MSC | 4 | 70 | 0.93 | 0.10 | 3.87 | 0.89 | 0.12 | 3.09 | Excellent |
| | Total C [g kg$^{-1}$] | 370-2500 / 5 | Abs., SG (m=2, w=21), MSC | 10 | 70 | 0.99 | 0.49 | 10.05 | 0.92 | 1.35 | 3.61 | Excellent |
| | pH | 350-2500 / 3 | Abs., SG (m=1, w=21), MSC | 7 | 70 | 0.93 | 0.12 | 3.81 | 0.85 | 0.18 | 2.62 | Accurate |
| C | SOC [g kg$^{-1}$] | 370-2480 / 1 | Abs., SNV, GSD (m=1, w=11, s=1) | 7 | 70 | 0.90 | 1.02 | 3.11 | 0.75 | 1.58 | 2.01 | Accurate |
| | POXC [g kg$^{-1}$] | 370-2440 / 3 | Refl., SG (m=2, w =21) | 7 | 70 | 0.93 | 0.03 | 3.80 | 0.79 | 0.05 | 2.22 | Accurate |
| | Total N [g kg$^{-1}$] | 350-2460 / 4 | Abs., SG (m=2, w=21), MSC | 7 | 70 | 0.97 | 0.05 | 5.87 | 0.88 | 0.09 | 2.93 | Accurate |
| | Total C [g kg$^{-1}$] | 390-2500 / 5 | Refl., SG (m=1, w=21), MSC | 9 | 70 | 0.96 | 0.98 | 5.36 | 0.91 | 1.54 | 3.40 | Excellent |
| | pH | 390-2500 / 5 | Abs., SG (m=2, w=21), MSC | 6 | 70 | 0.89 | 0.05 | 3.09 | 0.75 | 0.08 | 2.02 | Accurate |
| D | SOC [g kg$^{-1}$] | 390-2500 / 3 | Abs., SNV, GSD (m=2, w=21, s=1) | 6 | 70 | 0.97 | 0.81 | 6.01 | 0.95 | 1.05 | 4.64 | Excellent |
| | POXC [g kg$^{-1}$] | 390-2460 / 6 | Refl., SG (m=2, w=21) | 7 | 69 | 0.95 | 0.03 | 4.72 | 0.91 | 0.05 | 3.34 | Excellent |
| | Total N [g kg$^{-1}$] | 370-2500 / 4 | Abs., SG (m=2, w=21), MSC | 6 | 70 | 0.98 | 0.06 | 7.30 | 0.95 | 0.10 | 4.53 | Excellent |
| | Total C [g kg$^{-1}$] | 350-2500 / 2 | Refl., SNV, GSD (m=2, w=21, s=1) | 6 | 70 | 0.97 | 1.15 | 5.44 | 0.93 | 1.62 | 3.86 | Excellent |
| | pH | 350-2500 / 6 | Refl., SG (m=2, w=21), MSC | 9 | 70 | 0.99 | 0.06 | 9.79 | 0.95 | 0.13 | 4.48 | Excellent |
| E | SOC [g kg$^{-1}$] | 350-2500 / 3 | Abs., SG (m=1, w=25) | 3 | 53 | 0.82 | 1.25 | 2.35 | 0.78 | 1.35 | 2.17 | Accurate |
| | POXC [g kg$^{-1}$] | 350-2500 / 4 | Abs., GSD (m=2, w=21, s=21) | 4 | 53 | 0.82 | 0.05 | 2.41 | 0.81 | 0.05 | 2.31 | Accurate |
| | Total N [g kg$^{-1}$] | 350-2500 / 3 | Abs., SG (m=2, w=21) | 4 | 53 | 0.94 | 0.07 | 4.12 | 0.89 | 0.10 | 3.03 | Excellent |
| | Total C [g kg$^{-1}$] | 360-2500 / 3 | Refl., SG (m=1, w=21), MSC | 6 | 53 | 0.98 | 1.20 | 7.83 | 0.96 | 1.91 | 4.91 | Excellent |
| | pH | 350-2500 / 4 | Refl., SG (m=1, w=21), MSC | 7 | 53 | 0.98 | 0.10 | 7.15 | 0.94 | 0.17 | 4.10 | Excellent |
| F | SOC [g kg$^{-1}$] | 350-2500 / 3 | Refl., SG (m=1, w=21), MSC | 3 | 53 | 0.58 | 1.77 | 1.56 | 0.53 | 1.88 | 1.47 | Approximate |
| | POXC [g kg$^{-1}$] | 380-2500 / 2 | Refl., GSD (m =2, w=21, s=21) | 5 | 53 | 0.86 | 0.03 | 2.72 | 0.81 | 0.03 | 2.29 | Accurate |
| | Total N [g kg$^{-1}$] | 350-2500 / 3 | Abs., SG (m=1, w=21), MSC | 5 | 53 | 0.92 | 0.06 | 3.47 | 0.83 | 0.09 | 2.41 | Accurate |
| | Total C [g kg$^{-1}$] | 370-2500 / 6 | Abs., SG (m=1, w=21), MSC | 5 | 53 | 0.84 | 0.95 | 2.54 | 0.75 | 1.20 | 2.00 | Accurate |
| | pH | 370-2500 / 4 | Refl., SG (m=1, w=51) | 1 | 53 | 0.23 | 0.04 | 1.15 | 0.28 | 0.03 | 1.17 | Poor |
| All | SOC [g kg$^{-1}$] | 350-2500 / 3 | Refl., SNV, GSD (m=2, w=5, s=1) | 7 | 386 | 0.92 | 1.66 | 3.65 | 0.89 | 1.99 | 3.06 | Excellent |
| | POXC [g kg$^{-1}$] | 350-2500 / 1 | Refl., SG (m=1, w=21), MSC | 8 | 385 | 0.88 | 0.05 | 2.90 | 0.84 | 0.06 | 2.42 | Accurate |





| Total N [g kg⁻¹] | 350-2500 / 4 | Abs., SG (m=2, w=11) | | 7 | 386 | 0.92 | 0.14 | 3.53 | 0.88 | 0.17 | 2.91 | Accurate |
|---|---|---|---|---|---|---|---|---|---|---|---|---|
| Total C [g kg⁻¹] | 350-2500 / 2 | Abs., SNV, GSD (m=2, w=5, s=1) | | 6 | 386 | 0.96 | 2.31 | 5.04 | 0.94 | 2.93 | 3.97 | Excellent |
| pH | 350-2500 / 3 | Refl., SG (m=2, w=21), MSC | | 9 | 386 | 0.95 | 0.12 | 4.40 | 0.89 | 0.17 | 2.97 | Accurate |

**Table 3: Performance of local spectral models in the literature. All cited studies worked with lab measurements in the vis–NIR range on dried and sieved (< 2 mm) soil samples. The studies are numbered like in Fig. 7. The column pre-processing indicates if pre-processing was optimized for every soil property or not. Model types include partial least square regression (PLSR), cubist, neural network (NN) and support vector machine regression (SVM).**

| Field | Property | Area [ha] | N | Mean | CV [%] | R² | RMSE | RMSE [%] | RPD | RPIQ | Model type | Pre-processing | Evaluation | Depth [cm] | Carbonate |
|---|---|---|---|---|---|---|---|---|---|---|---|---|---|---|---|
| *1 Rodriguez-Febereiro et al. (2022)* | | | | | | | | | | | | | | | |
| | Total N [g kg⁻¹] | 29.4 | 96 | 1.9 | 36 | 0.69 | 0.3 | 16 | 1.89 | | PLSR | Optimized | Test-set | 0-40 | None |
| | SOM [g kg⁻¹] | | 96 | 43.9 | 47 | 0.73 | 10.6 | 24 | 1.94 | | PLSR | | | | | |
| *2 Camargo et al. (2022)* | | | | | | | | | | | | | | | |
| | SOC [g kg⁻¹] | 200 | 400 | 7.1 | 19 | 0.54 | 0.9 | 13 | 1.44 | | PLSR | Limited | Test-set | 0-40 | Unknown |
| *3 Sleep et al. (2022)* | | | | | | | | | | | | | | | |
| | pH (H₂O) | 124 | 85 | 6.05 | 13 | 0.49 | 0.78 | 13 | | | PLSR | Limited | Test-set | 0-10 | Present |
| | pH (CaCl₂) | | 85 | 5.46 | 20 | 0.48 | 0.85 | 16 | | | PLSR | | | | | |
| *4 Li et al. (2022)* | | | | | | | | | | | | | | | |
| | SOC [g kg⁻¹] | 600 | 562 | 7.9 | 65 | 0.81 | 3.8 | 48 | | 1.66 | Cubist | Limited | Test-set | 0-5.5; 13-17; 28-32; 58-62 | None |
| *5 Breure et al. (2022): 4 different fields with maximal distance of 10 km* | | | | | | | | | | | | | | | |
| | SOC [g kg⁻¹] | 37.7 | 97 | 13.0 | 26 | | 0.8 | 6 | | 7.4 | PLSR | Limited | Test-set | 0-25 | None |
| | pH | | 97 | 7.31 | 6 | | 0.12 | 2 | | 2.6 | PLSR | | | | | |
| *6 Alomar et al., 2021* | | | | | | | | | | | | | | | |
| | SOM [g kg⁻¹] | 32 | 143 | 20.8 | 27 | 0.87 | 2.1 | 10 | 2.60 | | PLSR | Optimized | Test-set | 0-20 | High |
| | Total N [g kg⁻¹] | | 150 | 1.6 | 26 | 0.84 | 0.2 | 12 | 2.11 | | PLSR | | | | | |
| *7 Singh et al. (2022)* | | | | | | | | | | | | | | | |
| BOKA-PAN | SOC [g kg⁻¹] | 1 | 128 | 9.9 | 99 | 0.97 | 1.6 | 16 | | 8.55 | NN | Optimized | Cross-validation | 0-10; 10-30; 30-60; 60-90 | Present |
| LAU-PAN | SOC [g kg⁻¹] | 1 | 123 | 13.8 | 73 | 0.96 | 2.1 | 15 | | 6.63 | NN | | | | | |
| TSA-CCI | SOC [g kg⁻¹] | 1 | 128 | 12.2 | 88 | 0.82 | 3.9 | 32 | | 3.2 | NN | | | | | |
| WAL-WIN | SOC [g kg⁻¹] | 1 | 128 | 19.4 | 52 | 0.96 | 0.3 | 2 | | 1.49 | PLSR | | | | | |
| General model | SOC [g kg⁻¹] | 4 | 507 | 13.8 | 78 | 0.96 | 4.2 | 30 | | 1.49 | NN | | | | | |
| *8 Riefolo et al. (2020)* | | | | | | | | | | | | | | | |
| | SOC [g kg⁻¹] | ca. 18 | 98 | 20.3 | 24 | 0.65 | 2.6 | 13 | | 2.37 | PLSR | Optimized | Cross-validation | 0-20 | High |
| *9 Kawamura et al. (2021)* | | | | | | | | | | | | | | | |
| | Total C [g kg⁻¹] | 40'000 | 162 | 30.5 | 56 | 0.94 | 4.9 | 16 | | 3.841 | PLSR | Limited | Test-set | 0-10 | Unknown |
| *10 Stafford et al. (2018)* | | | | | | | | | | | | | | | |
| Waikato | Total C [g kg⁻¹] | 81.3 | 172 | 34.1 | 81 | 0.95 | 6.4 | 19 | 4.33 | | PLSR | None | Cross-validation | 0-10; 10-20; 20-30; 30-40 | Very low |
| | Total N [g kg⁻¹] | 81.3 | 172 | 3.7 | 71 | 0.91 | 0.8 | 22 | 3.43 | | PLSR | | | | | |
| Canterbury | Total C [g kg⁻¹] | 151.9 | 168 | 20.5 | 63 | 0.91 | 4.0 | 20 | 3.41 | | PLSR | | | | | |
| | Total N [g kg⁻¹] | 151.9 | 168 | 2.2 | 59 | 0.92 | 0.4 | 18 | 3.57 | | PLSR | | | | | |
| *11 Seidel et al. (2019)* | | | | | | | | | | | | | | | |
| TS1 | SOC [g kg⁻¹] | Field-level | 120 | 10.1 | 47 | 0.97 | 0.9 | 9 | 5.11 | | PLSR | Optimized | Test-set | 0-5; 5-10; 10-20; 20-30; 30-60 | None |
| TS2 | SOC [g kg⁻¹] | Field-level | 90 | 9.1 | 54 | 0.92 | 1.4 | 15 | 3.21 | | PLSR | | | | | |
| *12 Nawar and Mouazen (2019)* | | | | | | | | | | | | | | | |
| Hessleskew | SOC [g kg⁻¹] | 12 | 122 | 20.7 | 18 | 0.6 | 2.0 | 10 | 1.72 | | PLSR | limited | Test-set | 0-20 | Present |
| Hagg | SOC [g kg⁻¹] | 21 | 139 | 18.7 | 18 | 0.75 | 1.7 | 9 | 2.05 | | PLSR | | | | | |
| *13 Vaudour et al. (2018)* | | | | | | | | | | | | | | | |
| | SOC [g kg⁻¹] | 6 | 146 | 10.7 | 48 | 0.81 | 2.2 | 21 | 2.31 | | PLSR | None | Cross-validation | Different depths | Present |
| | CaCO₃ [g kg⁻¹] | | 146 | 398.6 | 37 | 0.88 | 50.0 | 13 | 2.92 | | PLSR | | | | | |
| | Total N [g kg⁻¹] | | 146 | 0.8 | 44 | 0.92 | 0.1 | 13 | 3.57 | | PLSR | | | | | |
| | pH | | 146 | 8.62 | 2 | 0.9 | 0.06 | 1 | 3.11 | | PLSR | | | | | |
| *14 Gomez and Coulouma (2018): 8 fields with each 20 to 36 topsoil samples* | | | | | | | | | | | | | | | |
| General model | CaCO₃ [g kg⁻¹] | 9.3 | 240 | 209.4 | 87 | 0.97 | 29.8 | 14 | 6.06 | | PLSR | Limited | Cross-validation | topsoil | High |



|  | Property |  |  |  |  |  |  |  |  |  | Model |  |  |  |  |
|---|---|---|---|---|---|---|---|---|---|---|---|---|---|---|---|
| *15 Luca et al. (2017)* |  |  |  |  |  |  |  |  |  |  |  |  |  |  |  |
|  | Total C [g kg⁻¹] | 33 | 216 | 61.5 | 28 | 0.82 | 6.8 | 11 |  | 3.03 | SVM | Limited | Test-set | 0-20 | None |
| *16 Debaene et al. (2014)* |  |  |  |  |  |  |  |  |  |  |  |  |  |  |  |
|  | pH | 56 | 398 | 6.25 | 8 | 0.52 | 0.34 | 5 | 1.40 |  | PLSR | Limited | Test-set | 0-25 | Present |
|  | SOC [g kg⁻¹] |  | 398 | 11.5 | 20 | 0.72 | 1.2 | 10 | 2.00 |  |  |  |  |  |  |
| *17 Brodsky et al. (2013)* |  |  |  |  |  |  |  |  |  |  |  |  |  |  |  |
|  | SOC [g kg⁻¹] | 100 | 152 | 11.1 | 31 | 0.85 | 1.4 | 13 | 2.40 |  | PLSR | Limited | Test-set | 0-20 | Unknown |
| *18 Yang (2011)* |  |  |  |  |  |  |  |  |  |  |  |  |  |  |  |
|  | Total N [g kg⁻¹] | Farm-level | 122 | 2.0 | 29 | 0.92 | 0.2 | 8 | 3.63 |  | PLSR | Optimized | Test-set | Topsoil | None |
|  | SOC [g kg⁻¹] |  | 122 | 19.7 | 28 | 0.9 | 1.6 | 8 | 3.21 |  | PLSR |  |  |  |  |
| *19 Kuang and Mouazen (2011)* |  |  |  |  |  |  |  |  |  |  |  |  |  |  |  |
| Bramstrup Estate | Total C [g kg⁻¹] | Farm-level | 70 | 18.4 | 163 | 0.89 | 11.0 | 60 | 3.15 |  | PLSR | Limited | Test-set | 0-30 | Present |
|  | SOC [g kg⁻¹] |  | 70 | 17.4 | 144 | 0.96 | 6.2 | 36 | 4.95 |  |  |  |  |  |  |
|  | Total N [g kg⁻¹] |  | 70 | 1.7 | 118 | 0.93 | 0.6 | 35 | 3.88 |  |  |  |  |  |  |
|  | pH |  | 70 | 7.19 | 7 | 0.08 | 0.33 | 5 | 1.10 |  |  |  |  |  |  |
| Wimex | Total C [g kg⁻¹] | Farm-level | 128 | 16.5 | 73 | 0.74 | 7.0 | 42 | 2.00 |  |  |  |  |  |  |
|  | SOC [g kg⁻¹] |  | 128 | 13.8 | 51 | 0.75 | 3.0 | 22 | 2.00 |  |  |  |  |  |  |
|  | Total N [g kg⁻¹] |  | 128 | 1.3 | 77 | 0.75 | 0.3 | 23 | 2.10 |  |  |  |  |  |  |
|  | pH |  | 128 | 7.02 | 27 | 0.16 | 0.63 | 9 | 1.13 |  |  |  |  |  |  |
| Mepol Medlov | Total C [g kg⁻¹] | Farm-level | 205 | 15.3 | 13 | 0.14 | 2.2 | 14 | 0.95 |  |  |  |  |  |  |
|  | SOC [g kg⁻¹] |  | 205 | 14.8 | 14 | 0.12 | 1.9 | 13 | 1.07 |  |  |  |  |  |  |
|  | Total N [g kg⁻¹] |  | 205 | 1.7 | 14 | 0.09 | 0.3 | 18 | 0.81 |  |  |  |  |  |  |
|  | pH |  | 205 | 6.78 | 6 | 0.3 | 0.30 | 4 | 1.31 |  |  |  |  |  |  |
| General model | Total C [g kg⁻¹] | Three Farms | 408 | 16.3 | 92 | 0.83 | 6.6 | 40 | 2.45 |  |  |  |  |  |  |
|  | SOC [g kg⁻¹] |  | 408 | 15.0 | 80 | 0.83 | 5.4 | 36 | 2.49 |  |  |  |  |  |  |
|  | Total N [g kg⁻¹] |  | 408 | 1.6 | 63 | 0.79 | 0.5 | 31 | 2.21 |  |  |  |  |  |  |
|  | pH |  | 408 | 6.94 | 7 | 0.03 | 0.70 | 10 | 0.90 |  |  |  |  |  |  |
| *20 Wetterlind and Stenberg (2010)* |  |  |  |  |  |  |  |  |  |  |  |  |  |  |  |
| Bränneberg | SOC [g kg⁻¹] | 69 | 86 | 24.0 | 13 | 0.7 | 1.2 | 5 | 1.90 |  | PLSR | Limited | Test-set | 0-20 | Present |
|  | pH |  |  | 6.80 | 1 | 0.49 | 0.10 | 1 | 1.33 |  |  |  |  |  |  |
| Hacksta | SOC [g kg⁻¹] | 97 | 137 | 24.0 | 25 | 0.85 | 2.2 | 9 | 2.60 |  |  |  |  |  |  |
|  | pH |  |  | 6.60 | 2 | 0.33 | 0.19 | 3 | 1.10 |  |  |  |  |  |  |
| Kärrtorp | SOC [g kg⁻¹] | 62 | 89 | 49.0 | 35 | 0.71 | 5.3 | 11 | 1.50 |  |  |  |  |  |  |
|  | pH |  |  | 6.20 | 5 | 0.5 | 0.22 | 4 | 1.40 |  |  |  |  |  |  |
| Sjörstorp | SOC [g kg⁻¹] | 78 | 106 | 19.0 | 26 | 0.57 | 2.7 | 14 | 1.90 |  |  |  |  |  |  |
|  | pH |  |  | 6.90 | 6 | 0.48 | 0.31 | 4 | 1.40 |  |  |  |  |  |  |
| *21 Wetterlind et al. (2008)* |  |  |  |  |  |  |  |  |  |  |  |  |  |  |  |
|  | SOM [g kg⁻¹] | 97 | 50 | 41.0 | 27 | 0.89 | 3.2 | 8 | 3.00 |  | PLSR | Limited | Test-set | 0-20 | None |
| *22 McCarty and Reeves (2006)* |  |  |  |  |  |  |  |  |  |  |  |  |  |  |  |
|  | SOC [g kg⁻¹] | 20 | 544 | 11.0 | 45 | 0.88 | 1.6 | 15 | 3.06 |  | PLSR | Limited | Cross-validation | 0-10; 10-20 | None |
|  | Total N [g kg⁻¹] |  | 544 | 0.9 | 44 | 0.85 | 0.2 | 18 | 2.50 |  |  |  |  |  |  |
|  | pH |  | 544 | 6.00 | 8 | 0.53 | 0.31 | 5 | 1.45 |  |  |  |  |  |  |






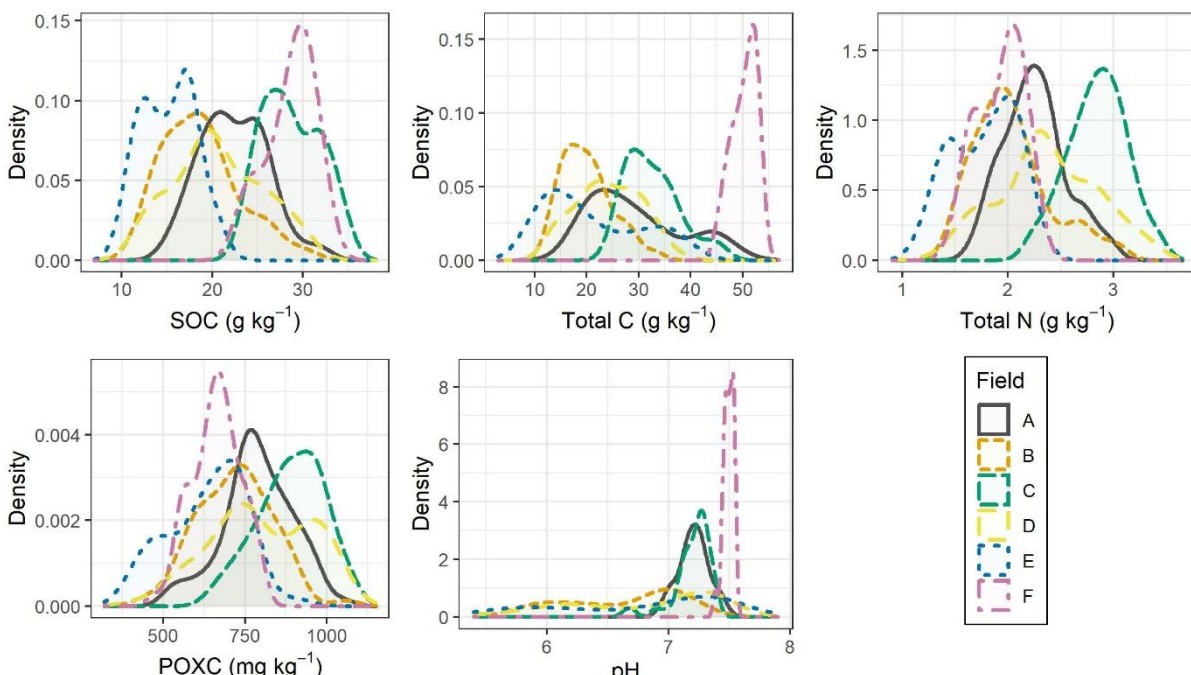

**Figure 1: Density plots of the reference samples for the five analyzed properties (SOC, total C, total N, POXC and pH). Field A to D contained each 70 samples and field E and F each 53 samples.**





**Figure 2: Biplots of principle component analysis with the first four principal components for the raw spectra and the pre-processed spectra according to the properties SOC, Total C, Total N, POXC and pH. The pre-processing is indicated in the figure and except for total N it was conducted on reflectance spectra (SG = Savitzky-Golay filter (m = order of derivative, w = window width), SNV = standard normal variate, GSD = gap segment derivative (m = derivative, w = window width, s = segment size), MSC = multiplicative scatter correction)**





**Figure 3: R², ratio of performance to deviation (RPD) and root mean square error (RMSE) calculated from the local models and field-specifically calculated from the general model for the six fields (A – F) and the five soil properties (SOC, total C, total N, POXC and pH). The overall RMSE of the general model is indicated with a black filled circle and the label "All". The calculated RMSE are compared with the error (mean standard error of 18 triplicates) of the lab measurements.**







**Figure 4: Variable importance in projection (VIP) for the local models of fields A to F and the general model that combined the datasets of all fields (All).**



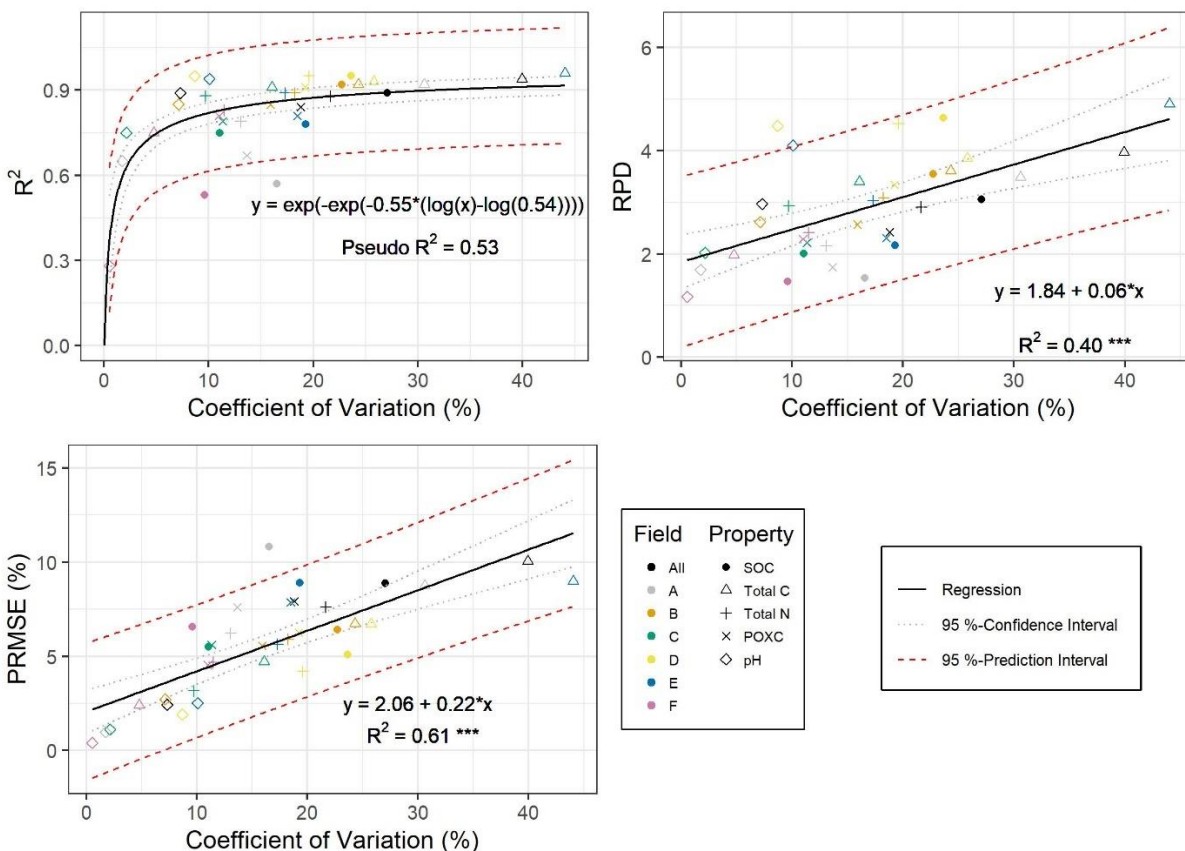

**Figure 5: R$^2$, ratio of performance to deviation (RPD) and percental root mean square error (PRMSE) in dependence of coefficient of variation for all spectral models (Fields A to F plus general model (All)). Significant codes for the linear regressions: *** < 0.001, ** < 0.01, *< 0.05. For R$^2$ a two-parameter Weibull function was fitted, and both fitted parameters were significant (p < 0.05). 95 % confidence and 95 % prediction intervals are indicated for all regressions**





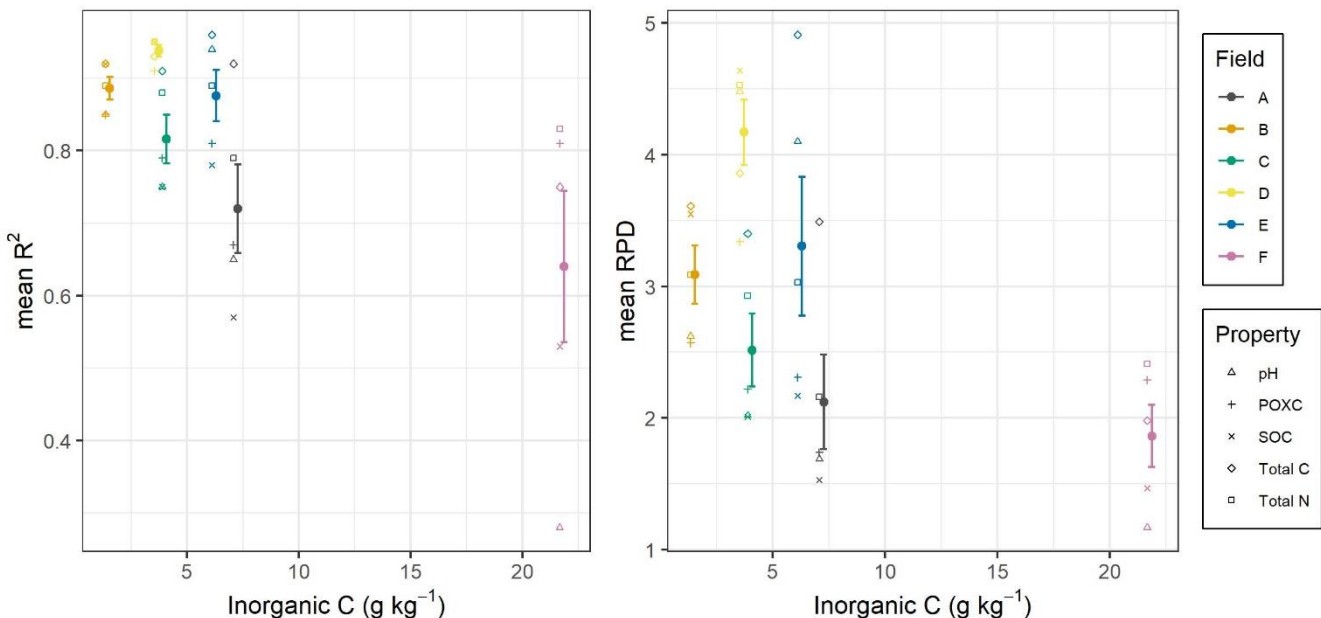

**Figure 6: R² and ratio of performance to deviation (RPD) from the local models for SOC, total C, total N, POXC, and pH aggregated (mean and standard error) per field (A-F) in dependence of mean inorganic C concentration of the field.**





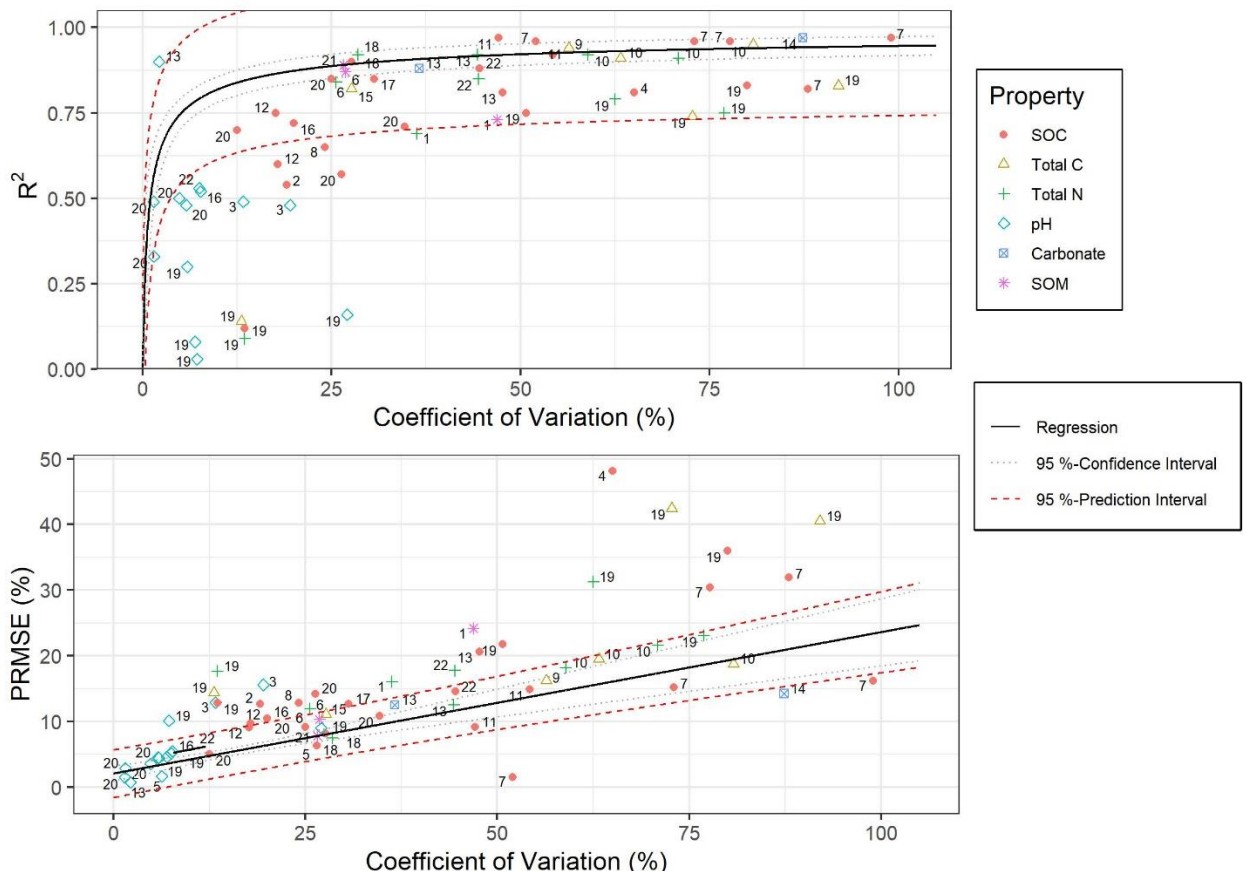

**Figure 7: R² and percental root mean square error (PRMSE) in dependence of coefficient of variation of 64 local models from 22**
**peer reviewed publications compared with the regressions found in this study. Only local models with a coefficient of variation <**
**105 % were plotted. The numbers of the points correspond to the numbered publications in Table 3.**

**Code and data availability:**

Data and R-codes are available on request.

**Author contributions:**

All co-authors conceptualized the paper. SO collected the data and conducted the analysis with the help of LS. SO wrote the

original draft and all co-authors participated in the writing and editing of the manuscript.

**Competing interests:**

The contact author has declared that none of the authors has any competing interests.




**Acknowledgements:**

The authors gratefully acknowledge Daniela Fischer and Patrick Neuhaus for their engaged support during the lab work. We also warmly thank Philipp Baumann for sharing his knowledge about the handling and modeling of spectral data. Lastly, we would like to thank the two anonymous reviewers for their constructive comments and suggestions.

**Financial support:**

This study was funded by the Land Systems and Sustainable Land Management Group from the Institute of Geography at the University Bern.