# Peer review of "Best performances of visible-near infrared models in soils with little carbonate - a field study in Switzerland"

_EGUsphere, 2023_

## Referee Comment (RC1)

**Comments to "Dataset variability and carbonate concentration influence the performance of local visible-near infrared spectral models"**

2023-07-25

**Contents**

**1 General comments**

**1.1 Research question and contribution**

The study tries to address the following research questions:

1. How do prediction errors of local spectral models (computed with samples from individual fields) differ from lab measurement errors?
2. "Does a general model computed [with data from all fields] improve the prediction on a target site with a poor local model performance?"
3. "What is the optimal variability in the dataset of a local model to achieve a good model performance?"
4. "Which field and soil characteristics (field size, soil texture, carbonate concentration) of the target site influence the performance of spectral models?"

The research questions (except question 3 if it is framed in terms of the variability of the target variable) are interesting, of methodological relevance — also in other disciplines than soil science — and fit well within the scope of SOIL. I like especially that the study synthesized data from many other studies, even though I think this analysis is misdirected. I also like that spectral modeling and validation seem to be of high quality and to have been conducted very thoughtfully (in particular because validation considers spatial autocorrelation). The manuscript is overall well written and clearly structured.

I have two major concerns:

1. **The analysis of how dataset variability controls model performance (research question 3) has major limitations and should be removed:** The first reason is that the analysis neglects that PRMSE, RPD, and $R^2$ are related to the CV simply by the way they are defined which means that we can learn much more about this relation through a simple simulation. The second reason is that measures of relative model performance (PRMSE, RPD, $R^2$) are of little practical relevance. The third reason is that model performance depends on the target variable identity, but the analysis neglects this and averages over all analyzed target variables.

2. **Data and code must be available to interpret, replicate and build upon the findings reported in the article and should be published**

I will discuss these points in the following sections.

**1.2 The analysis of how dataset variability controls model performance (research question 3) has major limitations and should be removed**

I completely agree that it is important to understand the controls of the predictive performance of spectral prediction models and also that it would be very useful if one could predict beforehand what predictive performance may be expected from such a model. However, I think that the analysis is misdirected for the following reasons (I use the same notation as you, i.e., RMSE is the RMSE computed on cross-validation or test data and $\text{RMSE}_{\text{cal}}$ is the same computed on calibration data):

1. The analysis of relations between model performance (PRMSE or RPD) and target variable variability (CV) does not appropriately consider that these variables are related due to the way they are defined: PRMSE and CV are always positively related (except in degenerate cases where the standard deviation of the data is 0 or the RMSE is 0, see R code 3 below, or — of course — due to noise) because both PRMSE and CV have the target variable mean ($\mu$) in the denominator. RPD and CV are always positively related when $\mu$ is positive (except in the degenerate cases where $\mu$ gets infinitely large or 0 or RMSE is 0, see R code 3 below) and negatively related when $\mu$ is negative (except in the degenerate cases where $\mu$ gets infinitely small or 0 or RMSE is 0). In my opinion, such a simple simulation tells us already more than an analysis of real data from actual model fits, which are also influenced by noise, and it can actually help to better

explain the results you have obtained with actual data. I will provide two examples, one for PRMSE and one for RPD:

- Example for PRMSE: The output of R code 3 shows that one can expect sudden breaks in the relation between PRMSE and CV for some combinations of RMSE, $\sigma$ (standard deviation of the target variable in the dataset), and $\mu$. Adapting the range to that observed for N (RMSE $\in (0.08, 0.3)$ (Fig. 3), $\sigma \in (0.2, 0.5)$ (Tab S1), $\mu \in (1.7, 2.9)$ (Tab S1)), one can see that the value ranges for N should not produce such breaks (See R code 4).

- Example for RPD: It is visible in Fig. 5, upper right panel (RPD vs CV), that the relation between RPD and CV differs between target variables — for instance, for pH, the slope RPD vs CV is steeper than for total C (whereas the CV range is smaller for pH and larger for total C). This can easily be explained by the small absolute RMSE for pH (see Fig. 3) which causes RPD to increase much stronger with CV than is the case for total C where a larger absolute RMSE (see Fig. 3) results in a smaller slope.

2. This also applies to $R^2$ because $R^2$ is directly related to the variance of the target variable: $R^2 = 1 - \frac{SS_{err}}{SS_{tot}}$ and $\sigma^2 = \frac{1}{n}SS_{tot}$ (See R code 3 below; $SS_{err}$ is the residual sum of squares of the model, $SS_{tot}$ the total sum of squares, $n$ is the sample size) and because $R^2$ is just a different way to express RPD: $R^2 = 1 - \frac{SS_{err}}{SS_{tot}} = 1 - \frac{n \cdot \text{RMSE}^2}{SS_{tot}} = 1 - \frac{n \cdot \text{RMSE}^2}{n \cdot \sigma^2} = 1 - \frac{1}{\text{RPD}^2}$. So the analysis for RPD and $R^2$ use the same variable, just transformed differently.

3. Model performance is strongly dependent on the target variable identity because the target variable identity defines the ranges for $\mu$, $\sigma$, and RMSE (for example RMSE for pH will hardly ever be larger than 14, whereas RMSE for POXC, for example, can easily be larger than 14). Point 1 in this section has shown that these ranges are strong controls on the relations between PRMSE and CV and RPD and CV (see for example the relation of RPD vs CV for pH, as described above). The analysis neglects target variable identity; the computed models simply average over values for different target variables. This causes large prediction uncertainties and shows that it does not make sense to compute models which average relative estimates for model performance over all target variables.

4. I think that measures of relative model performance (PRMSE, RPD, $R^2$) are of little practical relevance to users of spectral prediction models, in contrast to estimates of absolute model performance (e.g., RMSE) because only absolute estimates (and consequently absolute uncertainty estimates) can be used in subsequent analyses or decision processes (e.g., to decide whether to apply fertilizer and how much, or to model nutrient limitations of crop plants). Estimates of relative model performance are indeed useful to compare how the relative performance differs, but this use case and variable is interesting only to better understand *why* the models differ

in predictive accuracy, not as a standalone measure of predictive accuracy.

5. Assuming there are no measurement errors, the performance of a spectral prediction model will depend on how strongly spectral variables are related to the target variable, how strong interference by other spectral features is, and how much the model needs to extrapolate to make predictions for given samples.

Putting the previous points together, I think that in order to learn something about the controls of the predictive performance of spectral prediction models, we need to:
1. Focus on estimates of absolute predictive performance.
2. Consider target variable identity.
3. Understand the factors which cause correlations between spectral variables and a target variable and the factors which confound or mask these relations.
4. Understand what errors are caused by extrapolation.

I think that all of these are really hard problems which can impossibly be addressed by one single study alone and I do not suggest that you should or can address all above points using the dataset at hand and knowledge which is available today. In any case, I think that your study gets stronger if all analyses related to section 3.5 (and corresponding sections in the discussion: sections 4.4 and 4.5) are removed and research question 3 is removed.

Instead, I think it would be optionally much more useful to expand the analysis on which spectral properties or other sample properties control RMSE (currently section 3.6; this analysis exactly addresses the four points I think we need to address mentioned above). This analysis makes up only a small part of the results and discussion at the moment. I know how daunting it is if a reviewer requests additional analyses which probably could amount to another manuscript and I want to make clear that I suggest this only to provide constructive criticism: not only telling you what I think is misdirected and why, but how a useful direction would look like in my opinion. In light of this, please consider my suggestion to extend this analysis as optional.

**1.3 Data and code must be available to interpret, replicate and build upon the findings reported in the article and should be published**

I strongly encourage you to publish data and code of your analysis in a repository for the following reasons:

1. In the previous section, I argued that to better understand what controls model predictive performance, we need to understand the factors which cause correlations between spectral variables and a target variable and the factors which confound or mask these relations. Such an analysis will only have high enough accuracy if data from many local models are available, many more than can be provided by a single study.
2. The code (with package versions) is necessary to fully understand and reproduce (or replicate on new data) the spectral preprocessing and modeling.

3. Studies have repeatedly shown that unless data are published along the original publication, the probability that they won't be accessible is rather high (e.g., Tedersoo et al. (2021)).
4. There should be no legal reason prohibiting to publish the data and code (per the current data availability statement, p. 30).

**2  Specific comments**

1. l. 13 to 15: "… general models (combining all fields) for organic carbon, total carbon, total nitrogen, permanganate oxidizable carbon and pH using partial least squares regression. 24 out of 30 local models showed an accurate or even excellent performance (ratio of performance to deviation (RPD) > 2) …". I think the statement does not provide any useful information because it is not used to compare performance of models, but to make a standalone statement for one specific model (compare with point 4 in section "The analysis of how dataset variability controls model performance (research question 3) has major limitations and should be removed" of my review).

2. l. 15 to 16: "… and the root mean square errors (RMSE) of prediction were, except for pH, maximum five times higher than the lab measurement error." Whilst this statement is true (and RMSE is useful as a measure of model performance for standalone statements on model performance), I am not sure whether framing the comparison of model and lab RMSE only in terms of a ratio is useful (this also is the case in the results and discussion sections, for example) because one reason why the RMSE for pH is five times higher than the lab measurement error is that the pH lab measurement error is so small (see Fig. 3). Of course, all this makes only sense when one knows, what absolute error is tolerable.

3. l. 68 to 70: "However, it remains unclear what an optimal variability of a soil property in a project of local extent would be to achieve a high measurement accuracy in absolute values (low RMSE) but also relative to the range of the data (RPD)." Just to link this to my comments in section "The analysis of how dataset variability controls model performance (research question 3) has major limitations and should be removed": I think this question is misguided because it does not depend on the variability of the target variable how good a predictive model is (except that the variability of the target variable to predict should be larger than measurement errors), but on how well spectral variables are related to the target variable and masked by others. Of course, often a larger variability of the target variable implies more masking of spectral variables related to the target variable by other variables, but this larger variability of the target variable is not the cause.

4. l. 84: I think it makes sense to replace "measurement accuracy" by "measurement error".

5. l. 124 to 126: "To estimate the measurement error of SOC we took the

sum of the standard deviation of the total C measurement with the standard error of the inorganic C measurement because inorganic C measurement were for all samples done in triplicates." Shouldn't this rather be $\sqrt{\sigma_1^2 + \sigma_2^2}$ (instead of $\sigma_1 + \sigma_2$), where $\sigma_1$ is the standard deviation of the total C measurement and $\sigma_2$ the standard error of the inorganic C measurement?

6. l. 145 to 147: "The pre-processing techniques … led to around 100 meaningful combinations that were tested in model building and the final pre-processing option was selected based on the lowest RMSE." Have you selected the model with the lowest computed average RMSE or have you applied the one-sigma-rule also when comparing RMSE between different preprocessing workflows? If it's the former, redoing this analysis while applying the one-sigma-rule would make sense to avoid overfitting and to consider the uncertainty in the RMSE estimate when discussing which preprocessing methods performed well. In particular, I expect that within one standard deviation multiple preprocessing workflows performed equally well and one should decide that not enough information is available to pick prefer a specific one of those.

7. l. 149 to 151: "Since all soil properties showed a limited skewness (see Table S1 in the supplementary material) that was always in the range of -2 ≤ skew ≤ 2 which was proposed as acceptable to assume a normal univariate distribution (George and Mallery, 2010) we consider the application of PLSR appropriate, especially since it is robust to minor deviations from a normal distribution (Goodhue et al., 2012)." Some of the statements are not appropriate: (1) The skewness limits are a bad way to test whether a distribution is a normal distribution (See R code 1 below where I simulate data from a distribution which obviously is not normal and nevertheless has a skewness within the specified boundaries). (2) The data itself do not have to follow a normal distribution for PLSR to provide valid results (See R code 2 below where I simulate data from the same non-normal distribution and show that a PLSR model with good performance can be computed on these data).

8. l. 166: "… we calculated the mean Euclidean distances between all samples …". I assume the Euclidean distance was computed using the preprocessed spectral variables?

9. l. 174: "Since we used a cross-validation approach on the field scale, all models showed a very small bias." Please provide evidence for this claim, e.g. in Tab. 2 or in the supporting information.

10. l. 175 to 177: The sentence "RMSE was calculated according to Equation 1 where $\hat{y}_i$ is the prediction of the spectral model and $y_i$ the actual measured value in the laboratory." may be better understandable if it is replaced by (additions in bold): "RMSE was calculated according to Equation 1 where $\hat{y}_i$ is the prediction of the spectral model **for sample** $i$ and $y_i$ the actual measured value in the laboratory **for the same sample**."

11. l. 184 to 185: "… RPD is the best parameter to compare models of different scales." Actually, it is not, but it is one option to do so. For example, as described in point 2 of section "The analysis of how dataset variability controls model performance (research question 3) has major limitations and should be removed", RPD is just a transformation of $R^2$.

12. l. 204 to 205: "The VIP method can deal with multicollinearity and is therefore suitable for the interpretation of spectral models (Baumann et al., 2021)." Baumann et al., 2021 did not analyze to what extent VIP can deal with multicollinearity.

13. l. 217: I assume "independence" should be "in dependence".

14. l. 221 to 222: "… we analyzed the influence of mean carbonate concentrations, soil texture and field size on the model metrics but only the mean carbonate concentration showed effects and is therefore presented in the results." This description needs to be much more detailed: (1) How did you analyze this (regression, correlation, in case of regression, which distribution was assumed for the target variable? (2) Which models were computed (what variables were included)? (3) How (based on what criteria) were variables included/excluded? (4) Was this analysis performed separately for different target variables (as suggested in section "The analysis of how dataset variability controls model performance (research question 3) has major limitations and should be removed" above)? (5) Is the uncertainty in the estimated RMSE small enough to not affect your results and have you checked this?
In addition, please provide results for all models which were computed to support your claim that "only the mean carbonate concentration showed effects" (this can be shown in the supporting information).
If this is an option for you, I strongly encourage you to expand this analysis, perhaps combining this with the analysis of the variable importance, because this is in my opinion the right direction to better understand the controls of model predictive performance (compare with my comments in section "The analysis of how dataset variability controls model performance (research question 3) has major limitations and should be removed").

15. l. 225 to 226: Please specify the versions of the R packages as this can be important to reproduce your analyses.

16. l. 246: "There was no pre-processing combination that proved to be suitable for all models of the same field (Fig. S2 in the supplementary material) or of the same soil property (data not shown)." Please provide evidence for this claim, e.g. in the supporting information.

17. l. 268 to 269 and elsewhere: "The field-specifically calculated $R^2$". Maybe "field-specific $R^2$" is more concise?

18. l. 283 to 284: "bit higher for POXC and substantially higher for pH

(Fig. 3)." Do yo mean a bit higher for pH and substantially higher for POXC? The figure does not seem to fit to the text, at least if this difference is meant in terms of absolute values (~0.15 to 0.25 pH units versus >20 POXC units) (compare also to point 2 in this section of my review).

19. Section 3.5: Are uncertainties in the PRMSE and RPD negligible for the results of your analyses? (Please note my comments in section "The analysis of how dataset variability controls model performance (research question 3) has major limitations and should be removed" above. I just wanted to mention that this point is not clear).

20. All analyses of the relation of $R^2$, PRMSE, RPD versus some other variable: You've described that you used a Weibull distribution for $R^2$ since $R^2$ must be in [0, 1]. However, prediction intervals shown in Fig. 5 and 7 contain values larger than 1 and thus do not comply with this assumption. Similarly, PRMSE and RPD cannot have negative values, but prediction intervals for PRMSE in Fig. 5 and 7 contain negative values.

21.  l. 410 to 412: "Even though the local models of fields A and F had the lowest performance among all local models, they still showed, with exception of pH, approximate results which, depending on the research question might still provide useful information." I assume that "approximate" refers to the classification of RPD? In that case, I think the statement does not provide any useful information because it is not used to compare performance of models, but to make a standalone statement for one specific model (compare with point 4 in section "The analysis of how dataset variability controls model performance (research question 3) has major limitations and should be removed" of my review).

22.  l. 423 to 424: "… we did not find a better model performance with increasing mean clay content …" Please provide evidence for this claim, e.g. in the supporting information.

23.  l. 556: The URL "https://doi.org/10.2136/sssaj2002.6400" is wrong. It should be "https://doi.org/10.2136/sssaj2002.6400a".

24.  l. 698 to 699: "Lastly, we would like to thank the two anonymous reviewers for their constructive comments and suggestions." Thank you ;)

25. Fig. 1: Why are no density plots for carbonate concentration and soil texture variables shown here (or in the supporting information)?

26. Fig. 3: $R^2$, RMSE, RPD, and the lab measurement error are estimates and have uncertainties which should be added as error bars to the figures.

27. All figures (also in the supporting information) would benefit from (1) a higher image resolution, (2) larger legend keys, (3) thicker symbol lines or different symbol shapes such that it is possible to recognize better which points have which color.

**3 R code**

**3.1 R code 1**

Histogram of a non-normally distributed variable with -2 < skew < 2.

```
**simulate a distribution which is non-normal and nevertheless**
**has -2 < skew < 2**
library(moments)
set.seed(54)
y <- c(rnorm(50, 0, 2), rnorm(50, 30, 2))
hist(y, breaks = 30) # clearly non-normal
```

[Figure]

**Histogram of y**

```
moments::skewness(y)
```

> [1] -0.002923698

**3.2 R code 2**

PLSR can perform well also for a target variable which is non-normally distributed.

```
**show how PLSR performs on non-normal data**
library(pls)
library(tibble)
library(magrittr)

set.seed(8787)

**simulate data**
d1 <-
  tibble::tibble(
    x1 = c(rnorm(50, 0, 2), rnorm(50, 30, 2)),
    y = 5 * x1 + rnorm(100, 0, 1)
  ) %>%
  cbind(
    purrr::map_dfc(seq_len(100), function(i) {
      tibble::tibble(x = rnorm(100, 0, 2))
```

```
    }) %>%
      setNames(nm = paste0("x", seq_len(ncol(.)) + 1L))
  )

**show that y is non-normal**
hist(d1$y, breaks = 30)
```

[Figure]

**Histogram of d1$y**

```
**fit plsr model**
m1 <- pls::plsr(y ~ ., data = d1, scale = TRUE, center = TRUE, validation = "LOO")

**test on independent data**
d2 <-
  tibble::tibble(
    x1 = runif(100, 0, 30),
    y = 5 * x1 + rnorm(100, 0, 1)
  ) %>%
  cbind(
    purrr::map_dfc(seq_len(100), function(i) {
      tibble::tibble(x = rnorm(100, 0, 2))
    }) %>%
      setNames(nm = paste0("x", seq_len(ncol(.)) + 1L))
  )

**select number of latent variables**
ncomp <- selectNcomp(m1, method = "onesigma", plot = TRUE)

**"good" fit**
plot(d2$y ~ predict(m1, newdata = d2, ncomp = ncomp)[, , 1])
abline(0, 1)
```

[Figure]

**3.3 R code 3**

In each plot, columns differentiate RMSE levels and rows levels for $\sigma$.

```r
**simulate data to show that PRMSE, RPD, R^2 are always positively related**
**to CV (except in degenerate cases or for R^2, RPD, when mu is negative)**
library(ggplot2)
library(tibble)
library(magrittr)
library(dplyr)

**simulate data**
d <-
  expand.grid(
    mu = seq(-20, 20, length.out = 41),
    sd = seq(0, 200, length.out = 5),
    rmse = seq(0, 200, length.out = 5)
  ) %>%
  tibble::as_tibble() %>%
  dplyr::mutate(
    cv = sd/mu,
    prmse = rmse/mu,
    rpd = sd/rmse,
    r2 = 1 - rmse^2/sd^2
  ) %>%
  dplyr::filter(rmse <= sd) # exclude cases where the model is worse
                            # than an intercept identical to the sample average

**PRMSE vs CV**
d %>%
  dplyr::arrange(cv, prmse) %>%
  ggplot(aes(y = prmse, x = cv, group = paste0(sd, "_", rmse))) +
  geom_path() +
  facet_grid(sd ~ rmse)
```

[Figure]

```
**RPD vs CV**
d %>%
  dplyr::arrange(cv, rpd) %>%
  ggplot(aes(y = rpd, x = cv, group = paste0(mu), color = mu)) +
  geom_path() +
  facet_grid( ~ rmse)
```

[Figure]

```
**R2 vs CV**
d %>%
  dplyr::arrange(cv, r2) %>%
  ggplot(aes(y = r2, x = cv, group = paste0(mu), color = mu)) +
  geom_path() +
  facet_grid( ~ rmse)
```

[Figure]

**3.4  R code 4**

In the plot, columns differentiate RMSE levels and rows levels for $\sigma$.

```r
**same as r code 3, but for ranges covered for N in the real data**
library(ggplot2)
library(tibble)
library(magrittr)
library(dplyr)

d <-
  expand.grid(
    mu = seq(1.7, 2.9, length.out = 41),
    sd = seq(0.2, 0.5, length.out = 5),
    rmse = seq(0.08, 0.3, length.out = 5)
  ) %>%
  tibble::as_tibble() %>%
  dplyr::mutate(
    cv = sd/mu,
    prmse = rmse/mu,
    rpd = sd/rmse
  ) %>%
  dplyr::filter(rmse <= sd) # exclude cases where the model is worse
                            # than an intercept identical to the sample average

**PRMSE vs CV**
d %>%
  dplyr::arrange(cv, prmse) %>%
  ggplot(aes(y = prmse, x = cv, group = paste0(sd, "_", rmse))) +
  geom_path() +
  facet_grid(sd ~ rmse)
```

[Figure]

**References**

Tedersoo, Leho, Rainer Küngas, Ester Oras, Kajar Köster, Helen Eenmaa, Äli Leijen, Margus Pedaste, et al. 2021. "Data Sharing Practices and Data Availability Upon Request Differ Across Scientific Disciplines." *Scientific Data* 8 (1): 192. https://doi.org/10.1038/s41597-021-00981-0.

---

## Author Comment (AC1)

**Reply to Reviewer 1**

*12.09.2023*

**1  General answer**

We thank you for your detailed feedback and very constructive comments. We appreciate that you see value in our study and approved the handling and modelling of the spectral data.

You raised two major concerns and several specific comments that we address in our answers to your review. For more clarity between our answers and your comments, we have inserted your original statement in blue color.

We hope we addressed all comments to your satisfaction and thank you for your suggestions that help us to further develop the manuscript.

On behalf of all co-authors,

Simon Oberholzer

**2  Concern 1: The analysis of how dataset variability controls model performance (research question 3) has major limitations and should be removed**

We thank you for these detailed explanations which show the problematic of this analysis. We understand that our analysis about dataset variability and performance of spectral models has limitations because of the interrelation between coefficient of variation (CV) and the model performance parameters ($R^2$, RPD, PRMSE). We will remove research question 3 from the manuscript (chapter 3.5, 4.4 and 4.5) as you recommended. Moreover, we will put more focus on research question 4 about the controls of predictive performance of spectral models. We thank you for summarizing that topic in four bullet points:

1. Focus on estimates of absolute predictive performance.

2. Consider target variable identity.

3. Understand the factors which cause correlations between spectral variables and a target variable and the factors which confound or mask these relations.

4. Understand what errors are caused by extrapolation.

We will shift the focus from $R^2$ and RPD to RMSE, which also means that absolute model performance must be discussed in more detail for each property separately. As you suggested, we will expand the analysis to identify soil properties that influence the predictive model performance (section 3.5). So far, we showed the influence of mean carbonate content on model performance. We will expand this analysis and assess the impact of a) the variability of carbonate content and texture, and b) the correlations between soil characteristics (carbonate and texture) with the target variables on the performance (absolute prediction error) of the spectral models. This will provide insights into which factors mask the correlation between spectral variables and target variables.

**3   Concern 2: Data and code must be available to interpret, replicate and build upon the findings reported in the article and should be published.**

We will add a supplementary file containing the R codes for the spectral modelling where all steps can be followed.

**4   Specific comments**

**Comment 1:**

13 to 15: "… general models (combining all fields) for organic carbon, total carbon, total nitrogen, permanganate oxidizable carbon and pH using partial least squares regression. 24 out of 30 local models showed an accurate or even excellent performance (ratio of performance to deviation (RPD) > 2) …". I think the statement does not provide any useful information because it is not used to compare performance of models, but to make a standalone statement for one specific model (compare with point 4 in section "The analysis of how dataset variability controls model performance (research question 3) has major limitations and should be removed" of my review).

Since we have 6 local field models for each of the 5 properties and want to give a qualitative judgment about their relative performance, we use the RPD and the classification system proposed by Chang et al. (2001) and Zhang et al. (2018). Thereby we mainly want to give an overview about how many of the developed models showed a good performance. However, we agree with the you that the RPD should only be used when comparing multiple models of one property and will use it in such a context only. The RMSE as an absolute measure is the most important parameter and we will discuss it in more detail for the individual models, which can still be accurate even though they were classified as not satisfying based on RPD.

**Comment 2:**

15 to 16: "… and the root mean square errors (RMSE) of prediction were, except for pH, maximum five times higher than the lab measurement error." Whilst this statement is true (and RMSE is useful as a measure of model performance for standalone statements on model performance), I am not sure whether framing the comparison of model and lab RMSE only in terms of a ratio is useful (this also is the case in the results and discussion sections, for example) because one reason why the RMSE for pH is five times higher than the lab measurement error is that the pH lab measurement error is so small (see Fig. 3). Of course, all this makes only sense when one knows, what absolute error is tolerable.

In this statement we mainly want to show how much accuracy is lost when using local spectral models instead of conventional lab measurements. However, we agree that more detail about the lab measurement accuracy for each soil property is needed to make a fair comparison. Therefore, we will discuss the comparison between RMSE of the local models and the lab measurement error for each soil property separately. We will also discuss for each property what would be a tolerable prediction error.

**Comment 3:**

68 to 70: "However, it remains unclear what an optimal variability of a soil property in a project of local extent would be to achieve a high measurement accuracy in absolute values (low RMSE) but also

relative to the range of the data (RPD)." Just to link this to my comments in section "The analysis of how dataset variability controls model performance (research question 3) has major limitations and should be removed": I think this question is misguided because it does not depend on the variability of the target variable how good a predictive model is (except that the variability of the target variable to predict should be larger than measurement errors), but on how well spectral variables are related to the target variable and masked by others. Of course, often a larger variability of the target variable implies more masking of spectral variables related to the target variable by other variables, but this larger variability of the target variable is not the cause.

We will remove question 3 (see answer to Comment 1).

**Comment 4:**

84: I think it makes sense to replace "measurement accuracy" by "measurement error".

We thank you for the suggestion and will replace "accuracy" by "error".

**Comment 5:**

124 to 126: "To estimate the measurement error of SOC we took the sum of the standard deviation of the total C measurement with the standard error of the inorganic C measurement because inorganic C measurement were for all samples done in triplicates." Shouldn't this rather be $\sqrt{\sigma_1{}^2 + \sigma_2{}^2}$ (instead of $\sigma_1 + \sigma_2$), where $\sigma_1$ is the standard deviation of the total C measurement and $\sigma_2$ the standard error of the inorganic C measurement?"

We thank you for pointing this out. We will correct the calculation of the lab measurement error of SOC.

**Comment 6:**

145 to 147: "The pre-processing techniques … led to around 100 meaningful combinations that were tested in model building and the final pre-processing option was selected based on the lowest RMSE." Have you selected the model with the lowest computed average RMSE, or have you applied the one-sigma-rule also when comparing RMSE between different preprocessing workflows? If it's the former, redoing this analysis while applying the one-sigma-rule would make sense to avoid overfitting and to consider the uncertainty in the RMSE estimate when discussing which preprocessing methods performed well. In particular, I expect that within one standard deviation multiple preprocessing workflows performed equally well and one should decide that not enough information is available to pick prefer a specific one of those.

Yes, we did apply the one-sigma-rule when choosing the optimal preprocessing and you are right that many of the preprocessing methods worked similarly well. We will provide more details in the manuscript and add examples of model performance with different preprocessing workflows in the supplementary file.

**Comment 7:**

149 to 151: "Since all soil properties showed a limited skewness (see Table S1 in the supplementary material) that was always in the range of -2 ≤ skew ≤ 2 which was proposed as acceptable to assume a normal univariate distribution (George and Mallery, 2010) we consider the application of PLSR appropriate, especially since it is robust to minor deviations from a normal distribution (Goodhue et al., 2012)." Some of the statements are not appropriate: (1) The skewness limits are a bad way to test whether a distribution is a normal distribution (See R code 1 below where I simulate data from a

distribution which obviously is not normal and nevertheless has a skewness within the specified boundaries). (2) The data itself do not have to follow a normal distribution for PLSR to provide valid results (See R code 2 below where I simulate data from the same non-normal distribution and show that a PLSR model with good performance can be computed on these data).

We thank you for raising this important point and providing the example R codes. We will adapt this section accordingly and write that PLSR is more robust with normal distribution, but that normality is not a mandatory requirement.

**Comment 8:**

166: "… we calculated the mean Euclidean distances between all samples …". I assume the Euclidean distance was computed using the preprocessed spectral variables?"

We calculated the Euclidean distances using the raw spectra since in the beginning it was not clear which pre-processing method would be suitable. However, we agree that it makes sense to use the preprocessed spectra. We will adapt the distance calculations and use the preprocessed spectra, which we used most frequently for the trained models (Resampling interval = 3, Savitzky-Golay filter and multiplicative scatter correction)

**Comment 9:**

174: "Since we used a cross-validation approach on the field scale, all models showed a very small bias." Please provide evidence for this claim, e.g., in Tab. 2 or in the supporting information.

We will include the bias in Table 2 to support this statement.

**Comment 10 – 13:**

We agree on all these minor comments and will implement them as you suggested.

**Comment 14:**

221 to 222: "… we analyzed the influence of mean carbonate concentrations, soil texture and field size on the model metrics but only the mean carbonate concentration showed effects and is therefore presented in the results." This description needs to be much more detailed: (1) How did you analyze this (regression, correlation, in case of regression, which distribution was assumed for the target variable? (2) Which models were computed (what variables were included)? (3) How (based on what criteria) were variables included/excluded? (4) Was this analysis performed separately for different target variables (as suggested in section "The analysis of how dataset variability controls model performance (research question 3) has major limitations and should be removed" above)? (5) Is the uncertainty in the estimated RMSE small enough to not affect your results and have you checked this? In addition, please provide results for all models which were computed to support your claim that "only the mean carbonate concentration showed effects" (this can be shown in the supporting information). If this is an option for you, I strongly encourage you to expand this analysis, perhaps combining this with the analysis of the variable importance, because this is in my opinion the right direction to better understand the controls of model predictive performance (compare with my comments in section "The analysis of how dataset variability controls model performance (research question 3) has major limitations and should be removed").

We thank you for raising these important questions. With additional analyses we will investigate the effect of additional variables on the absolute model performance / RMSE. We will also provide more details on how variables were selected and how their correlation with the modeling errors was

assessed. Furthermore, we will add more information on variables, which did not show a significant impact on model errors and where therefore excluded from the manuscript. The stronger focus on the VIP values in the analyses will furthermore give insights on some potential interferences of soil properties in the vis–NIR range. As shown in Figure 4, for the models for each soil property, the important wavelengths are very site-specific, which can explain their varying model performances. We plan to check for correlations between soil properties (i.e., carbonate, texture) and spectral variables (as done for example by Conforti et al. (2018)) and compare them with the VIP analysis. This analysis might give some insights which soil properties might mask the spectral features of the target variables. Please find more details regarding this comment above (Concern 1) and answers to the five specific questions below:

1. Since we have six fields, we were always looking for linear relationships between site characteristics and parameters of model performance because it is very difficult to assume a different distribution with this number of locations. Thereby we mainly focused on linear correlations. We will add these important but missing details to the manuscript.
2. We explored linear relationships between field size, soil texture and carbonate content and model metrics (RMSE, $R^2$ and RPD). The manuscript will be completed with the information on these analyses.
3. We selected the available variables which we assumed had an impact on model performance.
4. We performed this analysis for all variables combined as well as for each target variable individually which we will specify in the manuscript.
5. We did not consider the uncertainty in RMSE for this analysis, but we will do include it in our revised version (as already suggested above; see Concern 1).

**Comment 15:**

225 to 226: Please specify the versions of the R packages as this can be important to reproduce your analyses.

We will indicate the R-package versions.

**Comment 16:**

246: "There was no pre-processing combination that proved to be suitable for all models of the same field (Fig. S2 in the supplementary material) or of the same soil property (data not shown)." Please provide evidence for this claim, e.g., in the supporting information.

We will give examples of model performance with different pre-processing methods in the supplementary material.

**Comment 17:**

268 to 269 and elsewhere: "The field-specifically calculated R2". Maybe "field-specific R2" is more concise?

We thank you for this suggestion. We will follow this advice.

**Comment 18:**

283 to 284: "bit higher for POXC and substantially higher for pH (Fig. 3)." Do you mean a bit higher for pH and substantially higher for POXC? The figure does not seem to fit to the text, at least if this difference is meant in terms of absolute values (~0.15 to 0.25 pH units versus >20 POXC units) (compare also to point 2 in this section of my review).

That was a writing mistake. We will correct it accordingly.

**Comment 19:**

Section 3.5: Are uncertainties in the PRMSE and RPD negligible for the results of your analyses? (Please note my comments in section "The analysis of how dataset variability controls model performance (research question 3) has major limitations and should be removed" above. I just wanted to mention that this point is not clear).

We will indicate the uncertainties of RMSE and RPD and take them into account when we report and discuss model performances.

**Comment 20:**

All analyses of the relation of R2, PRMSE, RPD versus some other variable: You've described that you used a Weibull distribution for R2 since R2 must be in [0, 1]. However, prediction intervals shown in Fig. 5 and 7 contain values larger than 1 and thus do not comply with this assumption. Similarly, PRMSE and RPD cannot have negative values, but prediction intervals for PRMSE in Fig. 5 and 7 contain negative values.

This comment concerns the analysis and figures about dataset variability and model performance that we will remove according to your recommendation.

**Comment 21:**

410 to 412: "Even though the local models of fields A and F had the lowest performance among all local models, they still showed, with exception of pH, approximate results which, depending on the research question might still provide useful information." I assume that "approximate" refers to the classification of RPD? In that case, I think the statement does not provide any useful information because it is not used to compare performance of models, but to make a standalone statement for one specific model (compare with point 4 in section "The analysis of how dataset variability controls model performance (research question 3) has major limitations and should be removed" of my review).

We thank you for the comment. We will, throughout the manuscript, use the RMSE for standalone statements and RPD for comparison between models (see above / Comment 1).

**Comment 22:**

423 to 424: "… we did not find a better model performance with increasing mean clay content …" Please provide evidence for this claim, e.g., in the supporting information.

We only looked at the linear correlation of the mean clay content with the model performance metrics, but we will expand this topic and provide more information on the methods as well (see above / Concern 1 and Comment 14).

**Comment 23:**

556: The URL "https://doi.org/10.2136/sssaj2002.6400" is wrong. It should be "https://doi.org/10.2136/sssaj2002.6400a".

We thank you for spotting this typo and will correct this DOI-link accordingly.

**Comment 24:**

698 to 699: "Lastly, we would like to thank the two anonymous reviewers for their constructive comments and suggestions." Thank you ;)

A sentence that was written to early but has proven to be very true.

**Comment 25:**

Fig. 1: Why are no density plots for carbonate concentration and soil texture variables shown here (or in the supporting information)?"

We agree with you, it would be helpful to have the density plots also for the carbonate contents and soil texture. We will include carbonate contents that in the figure as well. However, we only have texture measurements for composite samples per field analyzed with the standard sedimentation method. Additionally, we have soil texture data measured with laser diffraction method (LDM, Mastersizer 2000) in higher resolution (20 samples per field). The two methods are very comparable for sand content but the clay content was substantially lower and silt content higher with the LDM (also described in Ryzak and Bieganowski, 2011). We will scale the mean of the LDM measurements to the average of the sedimentation method while keeping the same relative standard deviation to obtain the heterogeneity of soil texture in the field. The higher resolution of soil texture will also be used to identify potential correlation with the target variables which might further show interferences with spectral features (see above; Concern 1, Comment 14).

**Comment 26:**

"Fig. 3: R2, RMSE, RPD, and the lab measurement error are estimates and have uncertainties which should be added as error bars to the figures."

We will add error bars for $R^2$, RPD and RMSE to the figures.

**Comment 27:**

"All figures (also in the supporting information) would benefit from (1) a higher image resolution, (2) larger legend keys, (3) thicker symbol lines or different symbol shapes such that it is possible to recognize better which points have which color."

We will enhance the resolution and the graphical presentation of the figures.

**5 References**

Chang, C. W., Laird, D. A., Mausbach, M. J., and Hurburgh, C. R., 2001. Near-infrared reflectance spectroscopy-principal components regression analyses of soil properties. Soil Science Society of America Journal 65, 480-490.

Conforti, M., Matteucci, G., and Buttafuoco, G., 2018. Using laboratory Vis-NIR spectroscopy for monitoring some forest soil properties. Journal of Soils and Sediments 18, 1009-1019.

Ryzak, M., and Bieganowski, A., 2011. Methodological aspects of determining soil particle-size distribution using the laser diffraction method. Journal of Plant Nutrition and Soil Science 174, 624-633.

Zhang, L., Yang, X. M., Drury, C., Chantigny, M., Gregorich, E., Miller, J., Bittman, S., Reynolds, W. D., and Yang, J. Y., 2018. Infrared spectroscopy estimation methods for water-dissolved carbon and amino sugars in diverse Canadian agricultural soils. Canadian Journal of Soil Science 98, 484-499.

---

## Author Comment (AC2)

**Reply to Reviewer 2**

*12.09.2023*

**General answer**

We thank you for your comments and recommendations. We numbered your comments and provided a specific answer for each one about how we will implement it in the manuscript.

We hope we addressed all comments to your satisfaction.

On behalf of all co-authors,

Simon Oberholzer

**Comment 1:**

"The paper is well-written and addresses a topical issue i.e., the scale of the data sets for optimizing spectral models. However, the authors rely on quality performance parameters that are not independent: RMSE, $R^2$ and RPD. Minasny et al. already demonstrated some years ago that there is an overlap between these parameters."

We thank you for pointing out this important concern. We completely agree and will remove this entire part from the manuscript (chapter 3.5, 4.4 and 4.5; see review 1). We will expand the analysis about how site characteristics (carbonate content, soil texture) influence the performance of local models and possibly cause interferences of spectral features (see also answer to review 1).

**Comment 2:**

In recent literature the RPD is mostly used for global comparison between studies. Bellon Maurel proposed to use the PIQ which is better suited for non-normally distributed data sets."

We decided to use RPD for two reasons:

1. Our data do not show major deviations from normal distribution which provides robust results with PLSR and also justifies the use of RPD.
2. Most readers are more familiar with RPD than RPIQ.

**Comment 3:**

Section 2.7. Regressions between RPD and CV are spurious, as the standard deviation appears in both terms. These relations should not be used (e.g., Fig. 5 and 7). The same applies to the relation between PRMSE and CV. Such relations always give a positive trend.

We will remove this analysis from the manuscript (see comment 1 and review 1).

**Comment 4:**

Finally, the authors conclude that local models perform better than general models, but do not really give a recommendation on a sampling strategy for fields/regions with no prior knowledge of the soil properties.

We thank you for pointing this out. Local models with vis–NIR show overall good performance, but a high carbonate content decreases the model performance. The question of how to sample a

completely unknown area is a crucial one in soil science and difficult to answer. For local vis–NIR projects, we have the following recommendations, that are probably not described clearly enough in the manuscript:

1) Check for the carbonate content in the soil and then one can decide whether vis–NIR calibration modeling is feasible (if you decide for the latter your sample size will increase substantially).
2) Think about the model accuracy (absolute prediction error) that can be achieved as described in our data. Is that prediction error tolerable or is it too high?

If we know what exactly causes the loss of precision in carbonate rich soils (what we will do in the next step) we might even be able to draw clearer conclusions.

**Comment 5:**

I am not a native speaker, but I would avoid using 'high' and 'low', e.g., small and large errors. Please check with a native speaker.

Thank you very much. We will check that and adapt the language accordingly.

**Comment 6:**

Generally, 'concentration' refers to solutions, while 'content' is a more general term that can also refer to a solid in another solid. You use both terms, but I would advise to systematically use 'content'.

We will use content instead of concentration throughout the manuscript. Thank you very much.

---

## Author Response (AR1)

**Resubmission of revised Manuscript:**

Original title: Dataset variability and carbonate concentration influence the performance of local visible-near infrared spectral models (egusphere-2023-1087)

**Dear Editor, dear Reviewers,**

Many thanks for your constructive and insightful comments on our manuscript. We revised the manuscript according to your comments as outlined in our review answer (12.09.2023). We are happy to resubmit an improved manuscript. As you suggested we removed research question 3 about the influence of dataset variability and expanded in exchange the analysis how field characteristics influence the performance of local models. Therefore, major changes in the manuscript text were necessary and we describe them in the following list. Additionally, a track-changed version of the manuscript is also provided.

We hope we addressed all comments to your satisfaction.
On behalf of all co-authors,

Simon Oberholzer

**List of relevant changes**

**Title**

Since we removed the variability part of the analysis, we changed the title to: "Best performances of visible-near infrared models in soils with little carbonate - a field study in Switzerland"

**Abstract**

The abstract was rewritten to put a stronger focus on absolute prediction performance (RMSE) and include the additional aspects of field characteristics that influence model performance (correlation between target variables).

**Introduction**

Descriptions of dataset variability in literature was removed as well as research question three and the former research question four became now number three.

**Methods**

Section titles (2.1 – 2.8) are still the same as in the first version of the manuscript. Since we included the variability in soil texture in our analysis, we describe the measurement in section 2.2. Section 2.7 "Assessment of site characteristics influencing model performance" was rewritten according to the expanded analysis.

30 **Results**

The old section 3.5 "Influence of dataset variability on model performance" was replaced with former section 3.6 "Site characteristics influencing model performance". The new section 3.5 contains three new figures (Fig. 6, 7 and 8) that were not part of the former manuscript. Additionally, Figure 4 was rearranged to group the VIP analysis per dataset and not per soil property like before. We discovered a small calculation error for estimating the model metrics in cross-validation. Therefore,

35 the model performance metrics in Table 2 slightly changed but did not influence the main results. The error was part of a function in the *simplerspec* R-package and has already been corrected by its creator Philipp Baumann.

**Discussion**

The sections 4.4 and 4.5 of the old manuscript were removed. The former section 4.6 became the new section 4.4 and was

40 expanded to three subsections: 4.4.1 Mean carbonate content, 4.4.2 Correlation between target variables, 4.4.5 Variability of clay content. The other discussion parts (4.1- 4.3) contain only minor changes.

**Conclusion**

We included the results of the expanded analysis also in the conclusion.

45

**References**

All the DOIs were carefully tested.

**Supplementary material**

50 Dataset descriptions of soil texture were included in Table S1. Table S2 is new and provides an example of pre-processing optimization on a specific dataset. Table S3 is also new compares to two methods that were used to measure soil texture. The Euclidean distances in Fig. S1 were recalculated with pre-processed spectra. Figures S3, S4, S5 and S6 are new and provide supplementary information about the expanded analysis how field characteristics influenced model performance.

55 **Data and code availability**

We decided to make the data and important R codes available upon publication on a zenodo repository.

---

## Referee Report (RR1)

**Comments to "Best performances of visible-near infrared models in soils with little carbonate - a field study in Switzerland"**

2023-12-29

**Contents**

**1 General comments**

You have improved the manuscript considerably and addressed nearly all comments I gave to the first version of the manuscript. I would like to thank you for the detailed comments and explanations. I have some comments on the expanded analysis how model predictive performance relates to field and soil properties. Of these, the first section (The analysis of how model performance relates to soil properties is not replicable and unreliable) mentions points I think should definitely be addressed.

My comments in subsections "The study should emphasize results for the RMSE in the analysis of how model performance relates to soil properties" and "The Interpretation of bad predictive accuracy for the models for field A and F should be clarified" are — I hope useful — suggestions to clarify or expand the presentation of results and their discussion. Finally, in section "Specific comments" I list specific comments about the current version.

My main concern with the new analysis how model performance relates to soil properties is that it is not replicable and not reliable because it is not defined what a "clear visual influence on model performance" (l. 198 to 199) is, how it

can be reliably identified, and because it is unclear how the uncertainty in the RMSE (respectively the $R^2$ or RPD) was considered in this analysis.

**1.1 The analysis how model performance relates to soil properties is not replicable and unreliable**

I think that the analysis how model performance relates to "field size, soil texture, carbonate content and with the correlation coefficients between SOC and total N in the dataset." (l. 198 to 199) is at the moment not replicable and unreliable, first because the reader is not told how to define a "clear visual influence on model performance" (l. 201 to 202), and second because it is unclear how the uncertainty in the RMSE (respectively the $R^2$ or RPD) was considered in this analysis. In particular:

1. l. 199 to 200: "With six local datasets as independent variables it is hardly possible to apply statistical tests and therefore, we relied on visible inspection."

   - This justification does not make much sense to me. First, it is possible to apply statistical tests even with few observations, even though the tests will probably show in at least some cases that more samples would be needed to analyze whether the relation is or is not compatible with some null hypothesis in a sensible way. This is exactly what good tests should do: tell us, when we are not allowed to draw conclusions because the uncertainties are too large.
   Second, a simple visual inspection is not convincing here: Why should we be licensed to give up estimating effect sizes and accept a visual inspection if sample sizes are small? Exactly in this case, formal procedures to estimate uncertainties are important to show the reader what may and what may not be concluded (and first of all how to arrive at a conclusion). Uncertainties may be large with only 6 fields, I agree, but we have to make a start somewhere.
   Please note that I do not imply that you should compute tests here, especially not for a null hypothesis of no effect, because I do not think this would make sense — no one would suggest that the variables you have analyzed are not at all related to model performance. I only suggest to remove this justification in favor of a visual inspection and to use a replicable procedure to analyze the data. Below, I make concrete suggestions for how to improve this analysis.

2. Regarding comment 14 of reviewer 1 (in your answers to reviewer 1, Oberholzer (2023)), you commented: "We did not consider the uncertainty in RMSE for this analysis, but we will do include it in our revised version (as already suggested above; see Concern 1)."

   - Currently, I can only see error bars for the RMSE in Fig. 6 and S3, however it is unclear how these uncertainties were considered in the analysis other than plotting error bars. This is also liked to the non-replicability of the visual analysis.

Instead, I suggest to estimate Pearson correlation coefficients or simple linear

regression models of the RMSE (see also the next subsection "The study should emphasize results for the RMSE in the analysis of how model performance relates to soil properties") versus field size, soil texture, carbonate content and with the correlation coefficients between SOC and total N in the dataset. This analysis should consider the uncertainties in the RMSE (e.g., via Monte Carlo analysis) and report the average correlations or slope with uncertainties.

You may use tests or other decision criteria to decide which relations to interpret in the main text, but you should clearly define them in the text and show all results at least in the supporting information (as is currently also the case).
This would give the reader more information about the uncertainties of this assessment, and about the effect of uncertainties in the estimated RMSE, and you would provide models one could test in the future (for example whether other studies find similar or higher correlations with other spectral measurement methods, etc.).

If you only provide a visual evaluation, you do not give opportunities for these comparisons and you do not show how big the influence of uncertainties in the RMSE are on the estimated relations and this makes the analysis unreliable.

**1.2 The study should emphasize results for the RMSE in the analysis of how model performance relates to soil properties**

Currently, when analyzing how model performance relates to "field size, soil texture, carbonate content and with the correlation coefficients between SOC and total N in the dataset." (l. 198 to 199), the manuscript de-emphasizes results for the RMSE at the expense of $R^2$ and RPD (for example, Fig. 5 only contains results for the relation of $R^2$ and RPD and results for the RMSE are shown fully only in the supporting information, Fig. S3). I suggest to emphasize results for the RMSE and not for $R^2$ and RPD because I think that the RMSE is the performance metric most interesting both for researchers and practitioners and because I do not agree with the justification you give for the de-emphaiss of results for the RMSE.

The justification you give for the de-emphasis of the RMSE in this analysis is that (l. 295 to 296) "… in absolute prediction performance (RMSE) we only found a clear effect for SOC (Fig. 6) and not for the other properties (Fig. S3 in the supplementary material)." As stated in the previous subsection, this analysis is not replicable and not reliable.
Looking at Fig. S3, I see that field A seems to have, in many cases, a distinct pattern in the RMSE versus soil properties than the models for the other fields (including the model for field F). For example, the model for SOC for field A has the largest RMSE (and largest uncertainty in RMSE) in comparison to the other models, despite having a carbonate content similar to the other fields, except field F. I suggest that the behavior of the model for field A in Fig. S3 is particularly interesting and warrants a better interpretation than currently given (see the next subsection "The Interpretation of bad predictive accuracy for the models for field A and F should be clarified").

Thus, I think that the relation of the RMSE to soil properties is more interesting

for researchers and practicioners and I think that absolute model performance offers more insights which have not been discussed sufficiently yet. For these reasons, I suggest to emphasize results for the RMSE. Please consider this as suggestion to clarify and improve the analysis because I think that currently, the analysis is less interesting (because it focuses on $R^2$ and RPD) and ambiguous (because I think the explanation why the models for field A often are less accurate should be a different one than why the models for field F often are less accurate, see the next subsection).

**1.3 The Interpretation of bad predictive accuracy for the models for field A and F should be clarified**

Your current analysis of the influence of carbonate concentrations on the predictive accuracy for the models for field A and F is in my opinion ambiguous. The manucsript states

- l. 293 to 295: "Fields A and F that showed lower model performance in terms of RPD had higher carbonate content, lower correlation coefficient between SOC and total N and higher variability in soil texture (compare also with density plots in Fig. 1)."

- l. 364: "We found an influence of carbonate content with lowest performance of local spectral models on fields A and F."

This can give the reader the impression that samples from fields A and F had a similar carbonate content, even though this is not the case. From Fig. 1 and Fig. S3, one can see that field A has an average carbonate content more similar to fields with good predictive performance (e.g. field E), but few samples with a high carbonate content. In contrast, field F has on average a high carbonate content and no (or very few) samples with lower carbonate content. All other fields do not seem to have samples with carbonate contents similarly large as in field F or A. Thus, a high average carbonate content does not seem to be necessary to cause bad predictive performance. It rather seems plausible that already few carbonate rich samples decrease model predictive accuracy.

Perhaps similar to you, I assume that the high variability in sample carbonate content (not the high average carbonate content per se) has caused the bad performance of prediction models for field A. Based on Fig. 1, one can see that only field A has such a large gradient in inorganic C content and despite average carbonate content similar to other fields with low carbonate content (for all samples), you have found a large RMSE and large uncertainty in the RMSE for field A (supporting Fig. S3). If you mean this, I suggest that you make clear that field A and field F differ in their properties and that it makes a difference whether the average carbonate concentration is high (field F) or whether the variability in carbonate content is high, even if there are only few samples with large carbonate content (field A): In the first case, at least total C content and pH value appear to be predictable quite accurately (with relatively low RMSE) (see Fig. S3), apparently because carbonate absorbance bands here can be used to explain total C content (which is mainly carbonate C), but carbonate bands cannot explain other soil properties related to SOM. In the latter case, only pH value appears to be predictable quite accurately (with relatively low RMSE)

because SOC and carbonate both contribute to total C, but carbonate bands and bands related to SOC interfere.

Here, it would be easy to test the assumption that already few samples with large carbonate content suffice to decrease predictive performance: Simply drop the samples with high carbonate content from field A and compute the prediction models (with same preprocessing settings) with the reduced dataset. If the resulting models have a much better predictive performance and you can exclude measurement artifacts, bad performance of model for field A is at least related to the high variability in carbonate content.

Similarly, you could include some samples with high carbonate content from field A or F to the datasets from the other fields and test whether this increases the RMSE of the corresponding prediction models. Finally, you could add few samples with low carbonate content to the dataset from field F to test this from the other side, i.e. whether high variability in carbonate content with few samples with low carbonate content also increases the RMSE of models (in comparison to the model with samples from field F only).

**2    Specific comments**

1.  l. 72 to 73: "Do field and soil characteristics (e. g. field size, soil texture, carbonate content, correlations of soil properties) of the target site influence the performance of spectral models?"

    - I would suggest to replace this by "How do field and soil characteristics (e. g. field size, soil texture, carbonate content, correlations of soil properties) of the target site relate to the performance of spectral models?" or split it into two parts, for example: "How does carbonate content influence the performance of spectral models?" (first question) and "How do other field and soil characteristics (e. g. field size, soil texture, correlations of soil properties) of the target site relate to the performance of spectral models?" (second question).
    The current wording implies that your analysis could *show* that there is a causal relation between specific properties of the target samples (or site), but since the study currently is observational, this is not the case. In some cases, one clearly can argue that specific soil properties have a causal influence on the predictive performance of the models, e.g. carbonate content. But your study does not *show* this, but uses information from previous studies on peaks caused by molecular structures in carbonates and results from your own analysis to elaborate how exactly carbonate content controls model performance (a causal factor which is already known/suggested in previous studies) and corelation between spectral variables and the target variables.
    For other properties, I am sceptical if there is any theory for how they should be causal (or how one should infer and define such causal effects based on observational data and available prior knowledge). For example, field size may be related to variability in soil properties

and thus spectral variability and variability in the relation between target variables and spectral variables for local fields. However, it is not hard to imagine that besides spatial variability, factors such as sampling design are important, too, and your analysis does not consider this in detail.

For these reasons, I would either remove causal wording or split the question into two parts, one where your study can elaborate causal relations (because we already know that they exist and roughly how they work), and one where exploratory analyses are interesting, but your current analysis does not provide causal information.

2. l. 223: "However, the RMSE of the local models for pH of fields A (0.08 $\pm$ 0.02) ...". This is the first occurrence of a mean $\pm$ error in the text an I suggest to state here, what this means, e.g. "However, the RMSE of the local models for pH of fields A (0.08 $\pm$ 0.02) (mean $\pm$ standard deviation) ...".

3. l. 295 to 296: "However, in absolute prediction performance (RMSE) we only found a clear effect for SOC (Fig. 6) and not for the other properties (Fig. S3 in the supplementary material)."

   - Since you used a non-replicable visual identification procedure, I cannot follow this conclusion. I suspect that some additional relations are "clear" when using a replicable identification procedure and even when considering uncertainty in the RMSE (e.g., with high probability, the Pearson correlation, even considering the uncertainty in RMSE, is larger than 0). I suspect this may be the case for example for the pH value.

4. l. 297 to 298: "We did not observe an influence of field size absolute contents of sand, silt and clay or variability of carbonate content on model performance (see Fig. S4 in the supplementary material)."

   - Again, it needs to be clarified how exactly (and reproducibly) the absence of an effect was identified.

5. l. 279 to 281: "It can clearly be seen that on field B and to a lower extent on field F, the same wavelengths were important in all soil properties related to soil organic matter (SOC, 280 total C, total N and POXC) ...". I would suggest to remove "clearly".

6. l. 366 to 368: "Looking at the correlation between spectral variables and inorganic C respectively SOC (Fig. 7) we can confirm this finding but have to add that on the local scale the absorption bands for carbonate and SOC varied substantially between different datasets."

   - Absorption bands (peaks) are caused at specific energy levels of the near infrared radiation because molecular bonds interact with the infrared radiation at specific energy levels. The position of specific absorption bands for carbonate and SOC is thus fixed. Do you

mean here that the relative intensity of "the absorption bands for carbonate and SOC varied substantially between different datasets"?

7. l. 374 to 376: "The higher lab measurement error with higher carbonate content can explain the lower model performance on soils with high carbonate content for SOC but not for the other four soil properties where model performance (in terms of RPD) still tended to be lower than on fields with little carbonate content (Fig. 5)."

   - Actually, you have not shown that "the higher lab measurement error with higher carbonate content *can explain* the lower model performance on soils with high carbonate content for SOC …" (my emphasis).
     One way to analyze this would be to simulate for datasets with low carbonate content, but similar range in SOC content, SOC content values from the measured SOC contents and the lab measurement error if the samples had a high carbonate content. Please note that I do not say you should conduct such an analysis, I just describe how one could analyze whether the lab measurement error for SOC content could have caused bad model performance for SOC content to provide constructive criticism, and state that without such an analysis, it is unlcear whether "the higher lab measurement error with higher carbonate content can explain the lower model performance on soils with high carbonate content for SOC …".

8. l. 389: "… makes it more difficult to attribute absorption features …". I suggest to replace "attribute" by "relate".

9. Tab. 2: Why do the $R^2$ values have no standard deviation for SOC and pH for field A?

10. Fig. 3: What do error bars for model RMSE represent?

11. Fig. 5: What do the filled circles and error bars represent in the figure? This does not seem to be explained in the legend or in the figure caption.

12. Fig. S5, field D: Variable POXC appears to have some missing values which results in `NA`s for all Pearson correlation coefficients involving this variable. Based on the plot, I assume that these are only very `NA` observations and I assume the values are missing at random. Thus, selection effects in omitting these samples are unlikely, and I recommend to recompute these correlations without the `NA` values.

**References**

Oberholzer, Simon. 2023. "Reply on RC1." Peer Review. https://doi.org/10.5194/egusphere-2023-1087-AC1.

---

## Author Response (AR2)

**Reply to Second Review from 29.12.2023 (minor revisions)**

*16.01.2024*

**1 General answer**

We thank you again for your careful evaluation of the manuscript and very constructive comments. You raised three general comments and 12 additional specific comments. For more clarity between our answers and your comments, we have inserted your original statement in blue color and gave a point-by-point reply.

We hope we addressed all comments to your satisfaction and thank you for your suggestions that helped us to further develop the manuscript.

On behalf of all co-authors,

Simon Oberholzer

**General comment 1: The analysis how model performance relates to soil properties is not replicable and unreliable**

We thank you for this comment. You raised the important point that we should apply some statistical analysis to make the analysis between model performance and soil properties more reliable and replicable. So far, we relied on a visual inspection because (as you also mentioned) the sample size (six fields) is for most statistical tests too small to formulate or even reject a null hypothesis. However, besides statistical tests you mentioned the option to report correlation coefficients between soil properties and model performance. As you suggested we did linear correlations with a Monte Carlo simulation (1000 iterations) to account for the uncertainty of the RMSE and reported in Figure 6 the found Pearson's correlation coefficient with standard deviation.

**General comment 2: The study should emphasize results for the RMSE in the analysis of how model performance relates to soil properties**

We agree that RMSE is for real practitioners probably the most interesting parameter. Therefore, we show now RMSE in dependence of soil characteristics for all five properties and not only for SOC in the main text. Moreover, we extend the discussion on the RMSE. Still, we are convinced that we still need $R2$ and RPD as one of our main discussion points, since we rely on them to compare the performances on the different fields, relatively to their soil property ranges (e.g. a pH model for a field with small pH values and/or narrow pH ranges will also lead to low RMSE, whereas the RPD relates the standard deviation of the pH values to the RMSE).

**General comment 3: The Interpretation of bad predictive accuracy for the models for field A and F should be clarified**

Thank you very much for this feedback. Indeed, field A is quite different from field F and so far, these differences have not been clearly stated in the manuscript. You suggested that for field F the lower model performance might have been provoked by the high carbonate content in all samples whereas for field A only a few samples with high carbonate content might have decreased model performance. As you suggested we calculated a new model for field A where we removed the samples with high carbonate content. We tried two thresholds (> 15 g kg$^{-1}$ and > 10 g kg$^{-1}$ inorganic C) but the new models showed very similar performance to the original model, and we decided therefore not report these reduced models in the discussion but instead stress the differences between field A and F more clearly. Similarly, we underlined that there are many more field and site characteristics, that can affect model performance, than only the ones we analyzed. We guess that the lower performance of field A might be caused by an additional factor, we do not have information about.

**2 Specific comments**

**Comment 1:**

72 to 73: "Do field and soil characteristics (e. g. field size, soil texture, carbonate content, correlations of soil properties) of the target site influence the performance of spectral models?"

• I would suggest to replace this by "How do field and soil characteristics (e. g. field size, soil texture, carbonate content, correlations of soil properties) of the target site relate to the performance of spectral models?" or split it into two parts, for example: "How does carbonate content influence the performance of spectral models?" (first question) and "How do other field and soil characteristics (e. g. field size, soil texture, correlations of soil properties) of the target site relate to the performance of spectral models?" (second question). The current wording implies that your analysis could show that there is a causal relation between specific properties of the target samples (or site), but since the study currently is observational, this is not the case. In some cases, one clearly can argue that specific soil properties have a causal influence on the predictive performance of the models, e.g. carbonate content. But your study does not show this, but uses information from previous studies on peaks caused by molecular structures in carbonates and results from your own analysis to elaborate how exactly carbonate content controls model performance (a causal factor which is already known/suggested in previous studies) and corelation between spectral variables and the target variables. For other properties, I am sceptical if there is any theory for how they should be causal (or how one should infer and define such causal effects based on observational data and available prior knowledge). For example, field size may be related to variability in soil properties and thus spectral variability and variability in the relation between target variables and spectral variables for local fields. However, it is not hard to imagine that besides spatial variability, factors such as sampling design are important, too, and your analysis does not consider this in detail. For these reasons, I would either remove causal wording or split the question into two parts, one where your study can elaborate causal relations (because we already know that they exist and roughly how they work), and one where exploratory analyses are interesting, but your current analysis does not provide causal information.

Thank you very much for this suggestion that makes the structure of the manuscript more concise. We decided to choose your first suggestion.

**Comment 2:**

223: "However, the RMSE of the local models for pH of fields A (0.08 ± 0.02) …". This is the first occurrence of a mean ± error in the text and I suggest to state here, what this means, e.g. "However, the RMSE of the local models for pH of fields A (0.08 ± 0.02) (mean ± standard deviation) …".

Thank your for this clarification. We changed the text as you suggested.

**Comment 3:**

295 to 296: "However, in absolute prediction performance (RMSE) we only found a clear effect for SOC (Fig. 6) and not for the other properties (Fig. S3 in the supplementary material)."

• Since you used a non-replicable visual identification procedure, I cannot follow this conclusion. I suspect that some additional relations are "clear" when using a replicable identification procedure and even when considering uncertainty in the RMSE (e.g., with high probability, the Pearson correlation, even considering the uncertainty in RMSE, is larger than 0). I suspect this may be the case for example for the pH value.

Now we calculated the correlation coefficients and can refer to them. We also address now the strong correlation between RMSE and carbonate content for pH which is basically caused by the very low variability in pH on fields with high carbonate content and therefore low RMSE.

**Comment 4:**

297 to 298: "We did not observe an influence of field size absolute contents of sand, silt and clay or variability of carbonate content on model performance (see Fig. S4 in the supplementary material)."

• Again, it needs to be clarified how exactly (and reproducibly) the absence of an effect was identified.

For RPD and $R^2$ we did not calculate correlation coefficients because they are not linear and with the little samples size, we do not want to assume a specific relationship. We changed the sentence and talk now that the three characteristics we chose, showed graphically most impact.

**Comment 5:**

279 to 281: "It can clearly be seen that on field B and to a lower extent on field F, the same wavelengths were important in all soil properties related to soil organic matter (SOC, 280 total C, total N and POXC) …". I would suggest to remove "clearly".

We removed "clearly" as you suggested.

**Comment 6:**

366 to 368: "Looking at the correlation between spectral variables and inorganic C respectively SOC (Fig. 7) we can confirm this finding but have to add that on the local scale the absorption bands for carbonate and SOC varied substantially between different datasets."

• Absorption bands (peaks) are caused at specific energy levels of the near infrared radiation because molecular bonds interact with the infrared radiation at specific energy levels. The position of specific absorption bands for carbonate and SOC is thus fixed. Do you mean here that the relative intensity of "the absorption bands for carbonate and SOC varied substantially between different datasets"?

Exactly. We enlarged the expression "absorption bands" to "relative intensity of absorption bands" as you suggested.

**Comment 7:**

374 to 376: "The higher lab measurement error with higher carbonate content can explain the lower model performance on soils with high carbonate content for SOC but not for the other four soil properties where model performance (in terms of RPD) still tended to be lower than on fields with little carbonate content (Fig. 5)."

• Actually, you have not shown that "the higher lab measurement error with higher carbonate content can explain the lower model performance on soils with high carbonate content for SOC ..." (my emphasis). One way to analyze this would be to simulate for datasets with low carbonate content, but similar range in SOC content, SOC content values from the measured SOC contents and the lab measurement error if the samples had a high carbonate content. Please note that I do not say you should conduct such an analysis, I just describe how one could analyze whether the lab measurement error for SOC content could have caused bad model performance for SOC content to provide constructive criticism, and state that without such an analysis, it is unclear whether "the higher lab measurement error with higher carbonate content can explain the lower model performance on soils with high carbonate content for SOC ...".

We changed the term "can explain" to "might be a possible explanation", to make clear that now statistical analysis was conducted.

**Comment 8:**

389: "... makes it more difficult to attribute absorption features ...". I suggest to replace "attribute" by "relate".

Thank you. Your suggestion is more concise, so we replaced "attribute" with "relate".

**Comment 9:**

Tab. 2: Why do the R2 values have no standard deviation for SOC and pH for field A?

Thank you for carefully reading the manuscript. We corrected that mistake and added the standard deviation.

**Comment 10:**

Fig. 3: What do error bars for model RMSE represent?

We added the meaning of the error bars in the figure caption ("The error bars for RMSE of spectral models represent standard deviations across the repeats in the cross-validation").

**Comment 11:**

Fig. 5: What do the filled circles and error bars represent in the figure? This does not seem to be explained in the legend or in the figure caption.

We added the explanation in the caption ("The error bars represent standard deviations across the repeats in the cross-validation"). As well we added this explanation in the caption of the figures in the supplementary material.

**Comment 12:**

Fig. S5, field D: Variable POXC appears to have some missing values which results in NAs for all Pearson correlation coefficients involving this variable. Based on the plot, I assume that these are only very NA observations and I assume the values are missing at random. Thus, selection effects in omitting these samples are unlikely, and I recommend to recompute these correlations without the NA values.

Thank you very much. Field D contains one NA for POXC that disturbed the calculation of correlation coefficients. We excluded this sample from the analysis for POXC and renewed the Figure S4.

---

## Author Response (AR3)

***Response after acceptance:***

Dear Editor,

Thank you very much for accepting our manuscript in SOIL. We updated the statement about data availability and inserted the DOI for the Zenodo repository, where the dataset and R-codes can be found. We also added the Zenodo repository in the reference list. Otherwise, we did not change the manuscript or the supplementary material.

Kind regards,

On behalf of all co-authors,

Simon Oberholzer